# Scaling Data-Driven Probabilistic Robustness Analysis for Semantic Segmentation Neural Networks

**Navid Hashemi**    **Samuel Sasaki**    **Ipek Oguz**    **Meiyi Ma**    **Taylor T. Johnson**
Department of Computer Science, Vanderbilt University, Nashville, TN 37235
{navid.hashemi, samuel.sasaki, ipek.oguz,
meiyi.ma, taylor.johnson}@vanderbilt.edu

## Abstract

Semantic segmentation neural networks (SSNs) are increasingly essential in high-stakes fields such as medical imaging, autonomous driving, and environmental monitoring, where robustness to input uncertainties and adversarial examples is crucial for ensuring safety and reliability. However, traditional probabilistic verification methods struggle to scale effectively with the size and depth of modern SSNs, especially when dealing with their high-dimensional, structured inputs/outputs. As the output dimension increases, these methods tend to become overly conservative, resulting in unnecessarily restrictive safety guarantees. In this work, we propose a probabilistic, data-driven verification algorithm that is architecture-agnostic and scalable, capable of handling the high-dimensional outputs of SSNs without introducing conservative and loose guarantees. We leverage efficient sampling-based reachability analysis to explore the space of possible outputs while maintaining computational feasibility. Our methodology is based on Conformal Inference (CI), which is known for its high data efficiency. However, CI tends to be overly conservative in high-dimensional spaces. To address this, in this paper, we introduce techniques to mitigate these sources of conservatism, enabling us to provide less conservative yet provable guarantees for SSNs. We validate our approach on large segmentation models applied to CamVid, OCTA-500 and Lung_Segmentation, and Cityscapes datasets, showing that it can offer reliable safety guarantees while lowering the conservatism inherent in traditional methods. We also provide a public GitHub repository[1] for this approach, to support reproducibility.

## 1   Introduction

Deep Neural Networks (DNNs) have demonstrated remarkable performance in numerous domains, including image classification, speech recognition, medical diagnosis, and autonomous systems. Despite their widespread adoption, DNNs remain highly vulnerable to input perturbations and adversarial examples—subtle changes to inputs that can drastically alter network outputs. This vulnerability raises serious concerns, particularly for safety-critical applications such as autonomous driving, medical imaging, and robotics, where incorrect predictions may lead to catastrophic outcomes.

The study of neural network robustness primarily falls into two categories: falsification-based methods, which focus on discovering adversarial examples [Madry et al., 2017, Goodfellow et al., 2014, Kurakin et al., 2016], and verification-based methods, which aim to formally certify a network's robustness under certain input perturbations [Zhou et al., 2024, Lemesle et al., 2024, Wu et al., 2024]. The former is useful for finding failure modes, but it lacks formal guarantees. On the other hand, the latter offers provable guarantees but often suffer from severe scalability limitations.

---

[1] https://github.com/Navidhashemicodes/SSN_Reach_ReLU_Surrogate

39th Conference on Neural Information Processing Systems (NeurIPS 2025).

Reachability analysis—computing the set of all possible outputs of a system given a bounded set of inputs—has emerged as a powerful tool for robustness verification. In neural networks, it can characterize how input perturbations propagate through the network and impact the final prediction. Nonetheless, reachability analysis is computationally intractable for large-scale networks or high-dimensional outputs. For instance, semantic segmentation networks (SSNs), which output pixel-level class labels, are particularly challenging due to their high-dimensional output spaces and complex architectures, often involving layers like transposed convolutions, max-pooling, and dilated convolutions. Traditional reachability-based methods struggle with such models, becoming overly conservative or failing to scale.

To address these limitations, probabilistic verification offers a compelling alternative [Fischer et al., 2021, Anani et al., 2024, Hao et al., 2022, Marzari et al., 2024]. Rather than verifying properties for all possible perturbations, it focuses on verifying them with high probability under a given distribution of input uncertainty. Probabilistic methods can scale better and provide meaningful statistical guarantees, especially when exact bounds or Lipschitz constants are unavailable or overly conservative. However, existing probabilistic approaches also face significant challenges: they often struggle with nontrivial output specifications, and fail to scale to large and deep networks like those used in semantic segmentation. In this work, we propose a scalable, architecture-agnostic, and low-conservatism probabilistic verification algorithm and introduce it for verifying semantic segmentation neural networks. Our method does not rely on restrictive assumptions about network structure or layer types and is well-suited to high-dimensional output spaces.

**Related Works**.

*Probabilistic verification of Neural Networks*. Neural networks are extremely nonlinear. This poses significant challenges for establishing probabilistic guarantees. Unlike deterministic techniques, the probabilistic methods for classification task do not directly extend to segmentation tasks. A key obstacle arises from the use of the union bound[2], which causes the guarantees to deteriorate rapidly in high-dimensional segmentation settings. A prominent research direction in probabilistic verification is randomized smoothing, first introduced for classifiers [Cohen et al., 2019] and later adapted to segmentation models [Fischer et al., 2021], with subsequent improvements in Anani et al. [2024], Hao et al. [2022]. More recently, conformal inference has also been explored as a way to provide probabilistic guarantees in classification tasks [Jeary et al., 2024]. In this work, we extend this line of research by applying conformal inference to the segmentation task. Although the guarantee formulation may differ, we numerically show that our method can significantly reduce the conservatism compared to existing SOTA on segmentation tasks.

*Deterministic verification of Neural Networks*. Deterministic verification of neural networks seeks to establish formal guarantees that a network's output satisfies a desired specification for all inputs within a certain set. This task has been approached through several established methods, including Duong et al. [2023], Katz et al. [2017], Anderson et al. [2020], Cheng et al. [2017], Tran et al. [2020a]. A large subset of these verification approaches can be interpreted within the broader scope of branch-and-bound algorithms [Bunel et al., 2020]. Leading tools in this category, such as $\alpha, \beta$-CROWN [Zhou et al., 2024], PyRAT [Lemesle et al., 2024], and Marabou [Wu et al., 2024], have shown strong performance in recent VNN-COMP challenges Brix et al. [2024]. Techniques like interval arithmetic [Cheng et al., 2017, Pulina and Tacchella, 2010] are also popular in the literature.

*Conformal Inference*. Conformal Inference has been also used for verification in a variety of applications. For instance, Lindemann et al. [2023], Zecchin et al. [2024] employ CI to guarantee safety in MPC control and Hashemi et al. [2023, 2024] employs CI for reachability of stochastic dynamical systems, see Lindemann et al. [2024] for a recent survey article.

## 2  Preliminaries

**Notations**. Boldface symbols ($\mathbf{X}$) represent **sets**, while calligraphic letters ($\mathcal{X}$) denote **distributions**. The expression $x \sim \mathcal{X}$ indicates that the random variable $x$ is drawn from the distribution $\mathcal{X}$, whereas $x \overset{\mathcal{X}}{\sim} \mathbf{X}$ means that $x$ is sampled from the set $\mathbf{X}$ according to $\mathcal{X}$. A **feedforward neural network** with input dimension $n_0$, $l$ hidden layers of sizes $n_1, \ldots, n_l$, and output dimension $N$, using the ReLU

---

[2]$\Pr[P_1 \wedge P_2 \wedge \ldots \wedge P_n] \geq 1 - \sum_{i=1}^{n} (1 - \Pr[P_i])$

activation function, is written as $[n_0, n_1, \ldots, n_l, N](\text{ReLU})$. For an input tensor $x \in \mathbb{R}^{h \times w \times nc}$, its **vectorized form** is given by $\mathbf{vec}(x) \in \mathbb{R}^{n_0}$, where $n_0 = nc \times w \times h$. We denote the **Minkowski sum** by $\oplus$ and the **ceiling operator** by $\lceil x \rceil$, which returns the smallest integer greater than or equal to $x \in \mathbb{R}$. Finally, an $\ell_2$ ball with radius $r \in \mathbb{R}^{n_0}$ centered at $x \in \mathbb{R}^{n_0}$ is denoted as $\mathbf{B}_r(x)$.

**Semantic Segmentation Neural Networks**  A *semantic segmentation neural network* (SSN) is a nonlinear function that assigns a class label to each pixel $x(i, j)$ in a multichannel input image $x$, producing a target class label $\text{mask}(i, j)$ from a predefined set of classes $\mathbf{L} = \{1, 2, \ldots, L\}$,

$$\text{SSN} : x \in \mathbb{R}^{h \times w \times nc} \mapsto \text{mask} \in \mathbf{L}^{h \times w},$$

where $h$, $w$, and $nc$ denote the height, width, and number of channels of the input image, respectively, and $(i, j) \in 1, \ldots, h \times 1, \ldots, w$ are pixel coordinates. In this paper, we denote the logits of SSNs by $y \in \mathbb{R}^{h \times w \times L}$, where the target class label $\text{mask}(i, j)$ is given by $\text{mask}(i, j) = \arg\max_{l \in \mathbf{L}} y(i, j, l)$.
Here, we also define the function $f : \mathbb{R}^{n_0} \mapsto \mathbb{R}^n$ as a nonlinear map between the flattened images and the flattened logits. Formally,

$$f : \mathbf{vec}(x) \in \mathbb{R}^{n_0} \mapsto \mathbf{vec}(y) \in \mathbb{R}^n, \quad n_0 = h \times w \times nc, \ n = h \times w \times L.$$

**Conformal Inference**  Consider a collection of i.i.d and positive scalar random variables $\mathbf{M} = \{R_1, R_2, \ldots, R_m\}$ sampled from $R \sim \mathcal{D}$, where $R_1 < R_2 < \ldots < R_m$. Given a new draw $R_{m+1}$ from the same distribution and a miscoverage level $\epsilon \in (0, 1)$, conformal inference (CI)[Vovk et al., 2005] constructs a prediction interval $C(R_{m+1}) = [0, d]$ such that $\Pr[\ R_{m+1} \in C(R_{m+1})\ ] \geq 1 - \epsilon$. In this paper, we refer to these scalar random variables as *nonconformity scores* and we refer to $\mathbf{M}$ as the calibration dataset. To compute the threshold $d$, we use the empirical distribution of nonconformity scores $R_i$ for $i \in 1, 2, \ldots, m$, and define a rank $\ell := \lceil (m + 1)(1 - \epsilon) \rceil$. A valid choice for the threshold $d$ is then $R_\ell$, the $\ell$-th smallest element of the augmented set $R_1, \ldots, R_m, \infty$, as defined in [Tibshirani et al., 2019], which yields the following marginal guarantee:

$$\Pr\left[R_{m+1} \leq R_\ell\right] \geq 1 - \epsilon, \tag{1}$$

where the probability is over the joint randomness of both the calibration and test samples [Tibshirani et al., 2019, Vovk et al., 2005]. The test sample $R_{m+1}$ typically refers to an unseen example drawn from the same distribution for which we aim to provide a coverage guarantee. In this paper, we also refer to this sample as the *unseen*, and denote it by $R^{\text{unseen}}$.

Given a sampled calibration dataset, the coverage level $\delta = \Pr[R^{\text{unseen}} \leq R_\ell]$ is itself a random variable following a $\text{Beta}(\ell, m + 1 - \ell)$ distribution [Angelopoulos and Bates, 2021]. The following equations shows the mean value and the variance of the beta distribution in terms of $m, \ell$:

$$\mathbf{E}\big[\delta\big] = \frac{\ell}{m + 1}, \quad \mathbf{Var}\big[\delta\big] = \frac{\ell(m + 1 - \ell)}{(m + 1)^2(m + 2)}, \tag{2}$$

that shows by appropriately tuning $m$ and $\ell$, we can significantly reduce the variance $\mathbf{Var}[\delta]$, to achieve tighter bounds. For instance, when $m = 8000$ and $\ell = 7999$, the variance of the corresponding Beta distribution is $3.123 \times 10^{-8}$, which is extremely small.

However, to present this in a completely formal way, we only consider low variance scenarios ($\mathbf{Var}(\delta) \ll 1$) and we also include this source of uncertainty by reformulating this guarantee with a double-step probabilistic guarantee. To include this uncertainty, we can utilize the cumulative density function, CDF, of beta distribution- a function known as the regularized incomplete beta function- and propose the following guarantee based on the definition of the CDF.

$$\Pr\left[\ \Pr[\ R^{\text{unseen}} \leq R_\ell\ ] > 1 - \epsilon\ \right] > 1 - \text{betacdf}_{1-\epsilon}(\ell, m + 1 - \ell). \tag{3}$$

where $\text{betacdf}_{1-\epsilon}(\ell, m + 1 - \ell)$ is the regularized incomplete beta function that is characterized by the choice of $m$ and $\ell$ evaluated at point $\delta_1 = 1 - \epsilon$. The one-step marginal guarantee in (1) holds only when the rank is set to $\ell := \lceil (m + 1)(1 - \epsilon) \rceil$. However, as shown in Appendix A, a key advantage of the double-step probabilistic guarantee is its flexibility: the strict relationship among $m, \ell$, and $\epsilon$ can be relaxed. Therefore, in this paper, we focus on the double-step guarantee and do not enforce the constraint $\ell := \lceil (m + 1)(1 - \epsilon) \rceil$. Instead, we tune $m, \ell$, and $\epsilon$ such that $\text{betacdf}_{1-\epsilon}(\ell, m + 1 - \ell)$

becomes negligibly small. Henceforth, for any logical statement $P(\ell, m) \in \{\text{true}, \text{false}\}$, defined over the hyper-parameters $m, \ell \leq m$, we refer to the double-step guarantee,

$$\Pr\left[\ \Pr\left[\ P(\ell, m)\ \right] > 1 - \epsilon\ \right] > 1 - \mathsf{betacdf}_{1-\epsilon}(\ell, m + 1 - \ell)$$

as the $\langle \epsilon, \ell, m \rangle$ guarantee and we say the statement $P(\ell, m)$ satisfies a $\langle \epsilon, \ell, m \rangle$ guarantee. We also refer to $\epsilon$ as the miscoverage level, $\delta_1 := 1 - \epsilon$, as the coverage level and $\delta_2 := 1 - \mathsf{betacdf}_{1-\epsilon}(\ell, m + 1 - \ell)$ as the confidence of guarantee. Appendix B provides a clarifying example on this topic.

**Deterministic and Probabilistic Reachability Analysis on Neural Networks**   Given a neural network $f : \mathbb{R}^{n_0} \to \mathbb{R}^n$ and an input set $\mathbf{I} \subset \mathbb{R}^{n_0}$, *deterministic reachability analysis* aims to construct a set $\mathbf{R}_f(\mathbf{I}) \subset \mathbb{R}^n$ such that, $x \in \mathbf{I}$ implies $f(x) \in \mathbf{R}_f(\mathbf{I})$.

In contrast, *probabilistic reachability analysis* assumes a distribution $\mathcal{W}$ over the input set $\mathbf{I}$, denoted $x \overset{\mathcal{W}}{\sim} \mathbf{I}$, and for a given miscoverage level $\epsilon$, proposes a set $\mathbf{R}_f^\epsilon(\mathcal{W}) \subset \mathbb{R}^n$ such that

$$x \overset{\mathcal{W}}{\sim} \mathbf{I} \quad \Rightarrow \quad \Pr\left[f(x) \in \mathbf{R}_f^\epsilon(\mathcal{W})\right] \geq 1 - \epsilon. \tag{4}$$

In this paper, we detail how to compute a probabilistic reach set with miscoverage level $\epsilon$, and sampling distribution $x \overset{\mathcal{W}}{\sim} \mathbf{I}$ by constructing a suitable calibration set $\mathbf{M}$ of size $m$, and selecting an appropriate rank $\ell$. Therefore, to keep the terminologies consistent, we reformulate the probabilistic reach set in (4) using the $\langle \epsilon, \ell, m \rangle$ guarantee, and we denote it by $\mathbf{R}_f^\epsilon(\mathcal{W} ; \ell, m)$, which satisfies

$$x \overset{\mathcal{W}}{\sim} \mathbf{I} \quad \Rightarrow \quad \Pr\left[\Pr\left[f(x) \in \mathbf{R}_f^\epsilon(\mathcal{W} ; \ell, m)\right] \geq 1 - \epsilon\right] \geq 1 - \mathsf{betacdf}_{1-\epsilon}(\ell, m + 1 - \ell).$$

**Adversarial Examples and Robustness of SSNs**   An *adversarial attack* on an image involves perturbing the input using a set of noise images $x_1^{\text{noise}}, \ldots, x_r^{\text{noise}}$ and corresponding coefficients $\lambda = [\lambda(1), \ldots, \lambda(r)]^\top$. These perturbations are applied through a parameterized function $\Delta_{\lambda, x^{\text{noise}}}(\cdot)$, generating the adversarial image as:

$$x^{\text{adv}} = \Delta_{\lambda, x^{\text{noise}}}(x) = x + \sum_{i=1}^{r} \lambda(i) x_i^{\text{noise}}. \tag{5}$$

We focus on evaluating the *robustness* of semantic segmentation networks (SSNs) under such adversarial settings, particularly under the unknown, bounded adversarial examples ($x^{\text{adv}} \in \mathbf{I}$) where the coefficients in vector $\lambda$ are unknown but bounded within specified ranges, i.e., $\lambda \in [\underline{\lambda}, \bar{\lambda}] \subset \mathbb{R}^r$.

A pixel $x(i, j)$ is deemed **attacked** if it faces perturbation. It is also considered **robust** if, for all adversarial examples, the predicted label remains unchanged: $\mathsf{SSN}(\Delta_{\lambda, x^{\text{noise}}}(x))(i, j) = \mathsf{SSN}(x)(i, j)$.

Consider another scenario where there exists perturbations such that the label at $(i, j)$ changes. In this case, the pixel is **non-robust** if for all perturbations we have $\mathsf{SSN}(\Delta_{\lambda, x^{\text{noise}}}(x))(i, j) \neq \mathsf{SSN}(x)(i, j)$. The pixel is also **unknown** if for some perturbations we have $\mathsf{SSN}(\Delta_{\lambda, x^{\text{noise}}}(x))(i, j) = \mathsf{SSN}(x)(i, j)$, and for some other perturbations we have $\mathsf{SSN}(\Delta_{\lambda, x^{\text{noise}}}(x))(i, j) \neq \mathsf{SSN}(x)(i, j)$.

To quantify robustness of SSN we consider two different metrics:

**Robustness Value**: The *Robustness Value (RV)* of a network is the percentage of pixels that remain robust under attack:   $RV = 100 \times (N_{\text{robust}} / N_{\text{pixels}})$,   where $N_{\text{pixels}} = h \times w$.

**Problem Formulation**   Let $x \mapsto \mathsf{SSN}(x)$ denote a semantic segmentation network:

**Problem 1**. The task involves classifying every pixel $x(i, j)$ into robust, non-robust, or unknown categories with respect to adversarial samples $x^{\text{adv}} \overset{\mathcal{W}}{\sim} \mathbf{I}$, ensuring compliance with a miscoverage level $\epsilon$ under strong probabilistic guarantees.

**Problem 2**. We aim to obtain the average robustness value $\overline{RV}$ across $K$ test images $\{x_1, \ldots, x_K\}$ evaluated against adversarial examples $x^{\text{adv}} \overset{\mathcal{W}}{\sim} \mathbf{I}$, ensuring validity under a miscoverage level $\epsilon$.

## 3   Scaling Probabilistic Reachability Analysis on SSN with Strong Guarantees

Given a specific distribution $x \overset{\mathcal{W}}{\sim} \mathbf{I}$ for sampling inputs $x$ from the input set $\mathbf{I}$, reasoning about the resulting output distribution $y \sim \mathcal{Y}$, is infeasible due to the high degree of nonlinearity in

neural networks. As a result, providing probabilistic coverage guarantees over the network's output space becomes a significant challenge. A key advantage of conformal inference is its distributional robustness—it produces valid guarantees that are robust to all family of distributions that can well represent the samples collected in the calibration dataset. This property makes conformal inference particularly well-suited for reachability analysis of neural networks, which are known for producing complex and often intractable output distributions.

However, a major challenge arises when applied to high-dimensional datasets: The scalar elements $R$ in calibration dataset are defined over the outputs $y \sim \mathcal{Y}$ and the space of all possible distributions $\mathcal{Y}$ that can adequately represent the calibration dataset is significantly larger in higher dimensions and thus conformal inference tends to be overly conservative. This poses a significant obstacle when applying CI to semantic segmentation networks (SSNs), which are known for producing extremely high-dimensional outputs. In this paper, we propose a novel learning-based neural network reachability analysis method for SSNs, that leverages deflative principal component analysis (PCA) [Mackey, 2008] to address the dimensionality challenges inherent in conformal inference when applied to provide coverage guarantees on neural network with high dimensional outputs.

In this section, we present our reachability technique in an incremental manner to clearly illustrate both the contributions and the challenges involved in applying conformal inference to SSNs. We begin by introducing a naive baseline approach, then highlight its limitations on high-dimensional spaces. Subsequently, we describe the enhancements we propose to overcome these challenges, culminating in the final version of our reachability algorithm.

### 3.1 Naive Reachability Technique via Conformal Inference

The first step to find a probabilistic reachset on neural networks with conformal inference (CI) is to generate a calibration dataset $\mathbf{M}$ of size $m$ and a training dataset $\mathbf{T}$ of size $t$.

**Generating train dataset.** We sample $t$ different inputs $x_j^{\text{train}}, j = 1, 2, \ldots, t$ from $\mathbf{I}$, with any distribution of interest, $x \overset{\mathcal{W}'}{\sim} \mathbf{I}$, and compute their corresponding outputs $\mathbf{vec}(y_j^{\text{train}}) = f(\mathbf{vec}(x_j^{\text{train}}))$. This gives us the training dataset, $\mathbf{T} = \left\{ (x_1^{\text{train}}, y_1^{\text{train}}), (x_2^{\text{train}}, y_2^{\text{train}}), \ldots, (x_t^{\text{train}}, y_t^{\text{train}}) \right\}$.

**Generating Calibration dataset.** We sample the input dataset with $m$ different inputs $x_i^{\text{calib}}, i = 1, 2, \ldots, m$ from the distribution $x \overset{\mathcal{W}}{\sim} \mathbf{I}$ and their corresponding set of outputs $\mathbf{vec}(y_i^{\text{calib}}) = f(\mathbf{vec}(x_i^{\text{calib}})), i = 1, 2, \ldots, m$. This gives us the input/output dataset $\mathbf{IO} = \left\{ (x_1^{\text{calib}}, y_1^{\text{calib}}), (x_2^{\text{calib}}, y_2^{\text{calib}}), \ldots, (x_m^{\text{calib}}, y_m^{\text{calib}}) \right\}$. We next attempt to use $\mathbf{IO}$ to compute a suitable nonconformity score, $R \in \mathbb{R}_{\geq 0}$. To that end, inspired by recent work on using conformal inference in time-series [Cleaveland et al., 2024], we design the nonconformity scores as follows in Equation (6). This design provides us a hyper-rectanglular set. Assuming the output $\mathbf{vec}(y) = [\mathbf{vec}(y)(1), \ldots, \mathbf{vec}(y)(n)] \in \mathbb{R}^n$, we propose, the nonconformity scores $R_i^{\text{calib}}$ as,

$$R_i^{\text{calib}} = \max \left( \frac{|\mathbf{vec}(y_i^{\text{calib}})(1) - c(1)|}{\tau_1}, \ldots, \frac{|\mathbf{vec}(y_i^{\text{calib}})(n) - c(n)|}{\tau_n} \right), \quad i = 1, 2, \ldots, m \quad (6)$$

where $c \in \mathbb{R}^n$ is the average of the members in the train dataset $\mathbf{T}$, and for $k = 1, 2, \ldots, n$, the normalization factor $\tau_k$ is proposed to normalize the elements of the random vectors $|\mathbf{vec}(y) - c|$ which can also be obtained using the train dataset as,

$$\tau_k := \max \left( \tau^*, \max \left( |\mathbf{vec}(y_1^{\text{train}})(k) - c(k)|, \ldots, |\mathbf{vec}(y_t^{\text{train}})(k) - c(k)| \right) \right),$$

where we set $\tau^* = 10^{-5} \left( \sum_{k=1}^{n} \sum_{j=1}^{t} |\mathbf{vec}(y_j^{\text{train}})(k) - c(k)| \right) / nt$, to avoid facing zero as our normalization factor. We then propose the calibration dataset as $\mathbf{M} = \left\{ R_1^{\text{calib}}, R_2^{\text{calib}}, \ldots, R_m^{\text{calib}} \right\}$.

The next step is to sort the nonconformity scores in $\mathbf{M}$ based on their magnitude. Without loss of generality let's assume $R_1^{\text{calib}} < R_2^{\text{calib}} < \ldots < R_m^{\text{calib}}$. Given a new draw $x^{\text{unseen}}$ from the distribution $x \overset{\mathcal{W}}{\sim} \mathbf{I}$ and the architecture of the neural network, the output $\mathbf{vec}(y^{\text{unseen}}) = f(\mathbf{vec}(x^{\text{unseen}}))$ follows a specific distribution. We call this distribution as $y \sim \mathcal{Y}$. Furthermore given the map we proposed to compute the nonconformity scores, the i.i.d random variables,

$$R^{\text{unseen}} = \max \left( \frac{|\mathbf{vec}(y^{\text{unseen}})(1) - c(1)|}{\tau_1}, \ldots, \frac{|\mathbf{vec}(y^{\text{unseen}})(n) - c(n)|}{\tau_n} \right) \quad (7)$$

adopt another distribution where we call it $R \sim \mathcal{D}$. Here $R^{\text{unseen}}$ and the elements $R_1^{\text{calib}}, R_2^{\text{calib}}, \ldots, R_m^{\text{calib}}$ are all sampled from the distribution $\mathcal{D}$. This means that we can apply conformal inference. Therefore, for the new draw $R^{\text{unseen}}$, given a desired rank, $\ell \leq m$ and a desired miscoverage level $\epsilon$, we can propose the following $\langle \epsilon, \ell, m \rangle$ guarantee,

$$\Pr \big[ \ \Pr[ \ R^{\text{unseen}} \leq R_\ell^{\text{calib}} \ ] > 1 - \epsilon \ \big] > 1 - \mathsf{betacdf}_{1-\epsilon}(\ell, m + 1 - \ell).$$

Following the definition in Equation (7), the logical statement $P_1(\ell, m) := [R^{\text{unseen}} \leq R_\ell^{\text{calib}}]$ is logically equivalent to

$$P_2(\ell, m) := \bigwedge_{k=1}^{n} \big[ c(k) - \tau_k R_\ell^{\text{calib}} \leq \mathbf{vec}(y^{\text{unseen}})(k) \leq c(k) + \tau_k R_\ell^{\text{calib}} \big],$$

which defines a hyper-rectangular set as the reachset for any unseen output of the neural network generated by $x \overset{\mathcal{W}}{\sim} \mathbf{I}$. Since $P_1(\ell, m)$ satisfies the $\langle \epsilon, \ell, m \rangle$ guarantee, this reachset also provides the same coverage guarantee on the unseen outputs of the neural network. Appendix C provides a clarifying example on this technique demonstrating that it is effective even on very deep neural networks, offering strong probabilistic guarantees with a reasonable number of required sampling. As the architecture of a deep neural network becomes more complex, the distribution of its output can exhibit increasingly intricate behavior. Conformal inference remains robust to any output distribution that represents the calibration dataset, allowing it to handle such complexities reliably. However, the tightness of the reachset does not persist when the dimensionality of the network's output increases.

- The first issue is that the naive technique can only produce a hyper-rectangular reachset, that can be a significant source of conservative in high-dimensional spaces.

- The second issue is that the space of distributions $\mathcal{Y}$ that can provide some distribution $\mathcal{D}$ which well represent the calibration dataset is significantly larger in high dimensional space.

## 3.2 From Hyper-Rectangular Reachsets to Generalized Zonotopic Representations

Starset [Bak and Duggirala, 2017] is an extension of zonotopes that we utilize here to represent our probabilistic reachsets. Our approach to constructing a general starset as the reachset is based on approximating the original deep neural network $f$ with a smaller and more computationally tractable ReLU network $g : \mathbb{R}^{n_0} \to \mathbb{R}^n$. The model $g$ is trained on the training dataset $\mathbf{T}$, which implies it is valid only for inputs $x$ sampled from the set $\mathbf{I}$.[3] The core idea is to apply deterministic reachability analysis to this simplified model using state-of-the-art tools such as NNV [Tran et al., 2020b], and then **inflate** its reach set to **account for approximation error** between $f$ and its surrogate $g$, thereby obtaining a probabilistic reachset for the original model $f$ with an $\langle \epsilon, \ell, m \rangle$ guarantee.

Training the surrogate model $g$ serves a **dual** purpose: it not only provides a better approximation for $f$, but also defines the shape of a **convex hull** which when inflated serves as a tighter over-approximation of the probabilistic reachset of $f$. Once $g$ is trained, this convex shape can be computed as a starset via deterministic reachability via NNV toolbox. To inflate this surrogate reachset, we follow the procedure outlined in the naive technique (Section 3.1) to generate a hyper-rectangle that bounds the prediction error with a provable $\langle \epsilon, \ell, m \rangle$ guarantee. In other words, we define $q(\mathbf{vec}(x)) = f(\mathbf{vec}(x)) - g(\mathbf{vec}(x))$ and apply the naive technique on $q(\mathbf{vec}(x))$ to compute its hyper-rectangular reachset with $\langle \epsilon, \ell, m \rangle$ guarantee[4]. We refer to this hyper-rectangle as *inflating set*.

In conclusion, if we denote the deterministic reachset of surrogate model as the starset $\mathcal{R}_g(\mathbf{I})$,

$$x \in \mathbf{I} \ \Rightarrow \ g(\mathbf{vec}(x)) \in \mathcal{R}_g(\mathbf{I})$$

and for some hyper-parameters $(\ell, m)$ and distribution $x \overset{\mathcal{W}}{\sim} \mathbf{I}$, the $\langle \epsilon, \ell, m \rangle$ guaranteed probabilistic reachset for prediction errors by $\mathcal{R}_q^\epsilon(\mathcal{W}; \ell, m)$,

$$x \overset{\mathcal{W}}{\sim} \mathbf{I} \ \Rightarrow \ \Pr \big[ \ \Pr[ \ q(\mathbf{vec}(x)) \in \mathcal{R}_q^\epsilon(\mathcal{W}; \ell, m) \ ] > 1 - \epsilon \ \big] > 1 - \mathsf{betacdf}_{1-\epsilon}(\ell, m + 1 - \ell),$$

---

[3] If the set $\mathbf{I}$ is very small, a linear map can be used as an alternative to the surrogate model $g$, since the input/output relationship of $f$ becomes easier to approximate in such cases.

[4] The naive technique can be viewed as a simplified version of the surrogate-based technique, where the training dataset is approximated by a simple average computation, ( $g(\mathbf{vec}(x)) = c$, $\mathcal{R}_g(\mathbf{I} = \{c\})$ ).

**Algorithm 1:** Learning-Based Principal Component Analysis through Deflation Algorithm

---

**Initialize** $z_1$ , $z_2$, $\ldots$, $z_t \leftarrow \mathbf{vec}(y_1^{\text{train}})$, $\mathbf{vec}(y_2^{\text{train}})$, $\ldots$, $\mathbf{vec}(y_t^{\text{train}})$ , $A \leftarrow [\,]$, $\underline{\mathcal{J}}$

**for** index $= 0, 1, \ldots, N-1$ **do**

    // Train the principal direction using stochastic gradient ascent

    $\vec{a}_{\text{index}} \leftarrow \arg\max\limits_{\vec{a}} \left[ \mathcal{J}(\vec{a}) := \frac{1}{t} \sum_{j=1}^{t} \vec{a}^\top z_j z_j^\top \vec{a} \right]$,     **s.t.**   $\|\vec{a}\|_2 = 1$

    $A.\mathbf{append}(\vec{a}_{\text{index}})$               // collect the principal direction in A

    **if** $\mathcal{J}(\vec{a}_{\text{index}}) < \underline{\mathcal{J}}$ **then**

        **break**                   // Stop if the variance falls below threshold.

     // Update the dataset by component removal along principal direction.

    $z_1$, $z_2$, $\ldots$, $z_t \leftarrow z_1 - (\vec{a}_{\text{index}}^\top z_1)\vec{a}_{\text{index}}$, $z_2 - (\vec{a}_{\text{index}}^\top z_2)\vec{a}_{\text{index}}$, $\ldots$, $z_t - (\vec{a}_{\text{index}}^\top z_t)\vec{a}_{\text{index}}$

---

then the probabilistic reachset of the model $f$ with the same $\langle \epsilon, \ell, m \rangle$ guarantee is obtainable by $\mathcal{R}_f^\epsilon(\mathcal{W}; \ell, m) = \mathcal{R}_g(\mathbf{I}) \oplus \mathcal{R}_q^\epsilon(\mathcal{W}; \ell, m)$, where $\oplus$ denotes the Minkowski sum,

$$x \overset{\mathcal{W}}{\sim} \mathbf{I} \;\Rightarrow\; \Pr\left[ \; \Pr[\; f(\mathbf{vec}(x)) \in \mathcal{R}_f^\epsilon(\mathcal{W}; \ell, m) \;] > 1 - \epsilon \;\right] > 1 - \mathsf{betacdf}_{1-\epsilon}(\ell, m + 1 - \ell).$$

This technique offers two key improvements over the naive approach presented in Section 3.1:

- The reachset is no longer constrained to a hyper-rectangle, eliminating the first source of conservatism inherent in the naive method.

- The calibration dataset is now defined using prediction errors rather than the network's raw outputs. Since prediction errors are typically of much smaller magnitude than the outputs of model $f$, this significantly reduces the conservatism of conformal inference in high-dimensional spaces.

Despite the improvements introduced in this section, the reachability analysis remains unscalable in high-dimensional settings. The core challenge is that training the surrogate model $g$—even when kept relatively small—often fails to converge its loss function to a good solution due to the high dimensionality of the output space. This implies that the magnitude of the prediction errors may not remain small relative to the output of the neural network. To mitigate this, we incorporate a scalable version of Principal Component Analysis (PCA), known as deflation algorithm, during the training process for $g$, a technique known to be effective for handling PCA in high-dimensional spaces. The next section provides a detailed explanation of this enhancement.

### 3.3 Training the Surrogate Model for SSNs via Principal Component Analyis

In this section, we present our methodologies to mitigate the dimensionality challenges to train the surrogate model $g : \mathbb{R}^{n_0} \mapsto \mathbb{R}^n$, and present our technique for the robustness analysis of SSNs.

In our robustness analysis of SSNs, as indicated by equation (5), adversarial perturbations are confined to an $r$-dimensional subspace of the input space, with $r \ll n_0$. Although input images are high-dimensional, the perturbations are not. Thus, we reformulate the surrogate as $g'(\lambda) : \mathbb{R}^r \mapsto \mathbb{R}^n$,

$$g'(\lambda) = g(\; \mathbf{vec}(\; x + \sum_{i=1}^{r} \lambda(i) x^{\text{noise}} \;) ) = g(\mathbf{vec}(x^{\text{adv}})) \quad \text{and} \quad \lambda \in [\underline{\lambda}, \overline{\lambda}] \subset \mathbb{R}^r,$$

and also the deterministic reachset $\mathbf{R}_g(\mathbf{I})$ based on the vector $\lambda$ as $\mathbf{R}_{g'}([\underline{\lambda}, \overline{\lambda}]) = \mathbf{R}_g(\mathbf{I})$.

**High-Dimensionality of Output Space**. To address this challenge, we divide the training process into two separate stages i.e., for a choice of $N \ll n$, we train for two models $g_1 : \mathbb{R}^N \to \mathbb{R}^n$ and $g_2 : \mathbb{R}^r \to \mathbb{R}^N$ and we define $g'(\lambda) = g_1(g_2(\lambda))$. The first stage trains $g_1$ and operates in a high-dimensional space; however, it is free of local optima, which ensures convergence to a global solution. The second stage trains the model $g_2$ within a non-convex optimization, but it operates in a low-dimensional space, making it easier to converge to a good solution.

The first stage performs dimensionality reduction. Let's sample training images $x_j^{\text{train}}$, for $j = 1, \ldots, t$, from the adversarial set $\mathbf{I}$, using sampling the coefficient vectors $\lambda_j^{\text{train}}$ form $[\underline{\lambda}, \overline{\lambda}]$. Then the

corresponding logits $\mathbf{vec}(y_j^{\text{train}})$ form a point cloud in $\mathbb{R}^n$. This stage aims to train the top $N$ principal directions of cloud, which are stored as columns of the matrix $A = [\vec{a}_0, \vec{a}_1, \ldots, \vec{a}_{N-1}] \in \mathbb{R}^{n \times N}$.

We utilize this matrix to represent the high dimensional logits $y_j^{\text{train}}$ with a lower dimensional representative $v_j \in \mathbb{R}^N$ where $v_j = A^\top \mathbf{vec}(y_j^{\text{train}})$. Principal Component Analysis is a technique to reduce the dimensionality, but it does not scale to high dimensional spaces. To address this issue, deflation algorithms have been proposed (see Algorithm 1) that compute principal components iteratively, in order of significance [Mackey, 2008]. At each iteration, once the most dominant principal component is identified, the dataset is projected onto the orthogonal complement of that direction—effectively removing its contribution—before computing the next principal direction.

In this paper, we adopt a learning based deflation algorithm that is presented in Algorithm 1. In this algorithm we set the components of the principal direction as trainable parameters and maximize the variance of the dataset in the direction of this vector. At each iteration of this algorithm, the optimization to train the principal direction have one global maxima one global minima and $n - 2$ saddle points, which $n$ is the dimension of logits $\mathbf{vec}(y_j^{\text{train}})$ [Mackey, 2008]. Therefore, although the training operates in a high-dimensional space, convergence to global maxima is guaranteed.

The second stage trains a relatively small $\mathrm{ReLU}$ neural network $g_2$ between the coefficient vectors $\lambda_j^{\text{train}}$ and the low dimensional representatives of logit $v_j, j = 1, \ldots, t$. This optimization operates in a low-dimensional space and can converge to a good solution. In conclusion we present the surrogate model for $f$ as $g(x^{\text{adv}}) = g'(\lambda) = Ag_2(\lambda)$ which is a scalable choice for SSNs.

**Detecting Robust, Non-robust and Unknown Pixels with Strong Probabilistic Guarantees**   Once the reachset over the SSN logits $y \in \mathbb{R}^n$ is constructed as a starset, we project it onto each logit component. This allows us to determine the range of values for each class label $l \in \mathbf{L}$ at every pixel location $(i, j) \in \{1, \ldots, h\} \times \{1, \ldots, w\}$. Consequently, each pixel $(i, j)$ is associated with $L$ logit intervals, one for each class. Let $l^* \in \mathbf{L}$ denote the class whose logit interval has the highest lower bound at pixel $(i, j)$. If this lower bound is not strictly greater than the upper bounds of all other classes' intervals, the pixel is labeled as *unknown* as it can obtain more than one labels under UBAA. In case it is greater, then if $l^* = \mathsf{SSN}(x)(i, j)$, the prediction at that pixel is labeled as *robust* and if $l^* \neq \mathsf{SSN}(x)(i, j)$, it is labeled as *non-robust*. The Algorithm 2 explains this procedure in detail.

# 4   Experiments

In our numerical evaluation, we pose four different research questions and address each of them using numerical results. The main RQs are available in this section and the rest of them are in Appendix G. We utilized a Linux machine, with 48 GB of GPU memory, 512 GB of RAM, and 112 CPUs.

**RQ1: How Does the Performance of our Methodology Compare to Existing Probabilistic Verification Methods for Neural Networks in the Literature?**   In this section, we present a comparison with the works of Fischer et al. [2021], Anani et al. [2024] on the Cityscapes dataset [Cordts et al., 2016]. Since the formulation of verification guarantees differs slightly across these approaches, we first provide a general overview of the techniques proposed in Fischer et al. [2021], Anani et al. [2024], followed by a brief description of our own method in Appendix D. This will clarify how the comparison can be meaningfully established.

**Comparison**.   The unknown pixels in our method are conceptually equivalent to the abstained pixels in Fischer et al. [2021], Anani et al. [2024]. Thus, given the same input set $\mathbf{B}_r(x)$ where $r$ is provided by Fischer et al. [2021], the comparison focuses to show which technique results in fewer uncertifiable mask pixels. To this end, we replicate the setup of Table 1 from Anani et al. [2024], using the same HrNetV2 model [Wang et al., 2020]trained on the Cityscapes dataset with the HrNetV2-W48 backbone[5]. Our findings are summarized in Table 1, which shows the average over 200 test images. Please note that the much lower rate of unknown mask pixels reported for our

---

[5]This model is considered large with 65,859,379 parameters, an input space of dimension $1024 \times 2048 \times 3$ and an output space of dimension $256 \times 512 \times 19$

Table 1: Percentage of uncertifiable pixels under different $(\sigma, \kappa, r)$ settings for Fischer et al. [2021], Anani et al. [2024], and our Naive approaches. Here $\delta_2 = 0.999$ in all experiments. See Appendix D for additional details on the parameters $(\sigma, \kappa, r)$. Our verification runtime is 210 seconds, and we set $\delta_1 = 0.99$ and $\mathcal{W}$ to be a uniform distribution, for all our CI based experiments.

| $\sigma$ | $\kappa$ | $r$ | ([Fischer et al., 2021]%) | [Anani et al., 2024] (%) | Ours (%) |
|---|---|---|---|---|---|
| 0.25 | 0.75 | 0.1686 | 7 | 5 | 0.0642 |
| 0.33 | 0.75 | 0.2226 | 14 | 10 | 0.0676 |
| 0.50 | 0.75 | 0.3372 | 26 | 15 | 0.0705 |
| 0.25 | 0.95 | 0.4112 | 12 | 9 | 0.0727 |
| 0.33 | 0.95 | 0.5428 | 22 | 18 | 0.0732 |
| 0.50 | 0.95 | 0.8224 | 39 | 28 | 0.0758 |

Table 2: Illustrates the model configurations and probabilistic guarantees. For each experiment, given hyperparameters $\ell, m$, and coverage $\delta_1 = 1 - \epsilon$, the confidence $\delta_2$ is computed. The reported verification runtime is averaged over all experiments and mainly depends on inference time and hyperparameters $(\ell, m)$, with minor sensitivity to perturbation magnitude and dimension $(e, r)$. Per-experiment runtimes are shown in Figure 7 of Appendix G.

| Dataset name | Model name | Input dimension | Output dimension | Number of Parameters | $(m, m-\ell)$ | coverage $\delta_1$ | confidence $\delta_2$ | Average runtime |
|---|---|---|---|---|---|---|---|---|
| M2NIST | m2nist_dc | $64 \times 84 \times 1$ | $64 \times 84 \times 11$ | 6,788,107 | $(1e5, 1)$ | 0.9999 | 0.9995 | 5.4 minutes |
| M2NIST | m2nist_ap_dc | $64 \times 84 \times 1$ | $64 \times 84 \times 11$ | 4,579,723 | $(1e5, 1)$ | 0.9999 | 0.9995 | 3.8 minutes |
| M2NIST | m2nist_ap_tc | $64 \times 84 \times 1$ | $64 \times 84 \times 11$ | 11,664,779 | $(1e5, 1)$ | 0.9999 | 0.9995 | 4.4 minutes |
| Lung Segmentation | UNet1 | $512 \times 512 \times 1$ | $512 \times 512 \times 1$ | 14,779,841 | $(8e3, 1)$ | 0.999 | 0.997 | 4 minutes |
| OCTA-500 | UNet2 | $304 \times 304 \times 1$ | $304 \times 304 \times 1$ | 5,478,785 | $(8e3, 1)$ | 0.999 | 0.997 | 3.75 minutes |
| CamVid | BiSeNet | $720 \times 960 \times 3$ | $720 \times 960 \times 12$ | 12,511,084 | $(8e3, 1)$ | 0.999 | 0.997 | 9 minutes |

method stems from considering a prior distribution. Thus we do not view this as an advantage, but rather as part of a trade-off between the two approaches, not a sign of superiority[6].

**RQ2 (1$^{\text{st}}$ Ablation Study): How does the technique scale with model size, number of perturbed pixels, and perturbation level?** To assess scalability, we test our surrogate-based verification method on an $r$-dimensional darkening adversary (see Figure 1), where $r'$ pixels in an image $x$ with intensity above $150/255$ in all channels are randomly selected for perturbation ($r = nc \times r'$). Each direction $x_i^{\text{noise}}$ corresponds to darkening a channel of one such pixel, and $x^{\text{adv}}$ is parameterized by an $r$-dimensional coefficient vector $\lambda \in [\underline{\lambda}, \overline{\lambda}]$. Here, $\overline{\lambda}$ induces full darkening (intensity zero), while $\underline{\lambda}$ applies partial darkening. This bound defines the perturbation space $x^{\text{adv}} \in \mathbf{I}$. The main motivation for this choice of threat model is that it offers greater flexibility to adjust the sparsity and magnitude of perturbations, allowing for a more thorough and detailed evaluation of our method's performance.

Per request of RQ2, in these experiments, we studied high-dimensional datasets (e.g., Lung Segmentation [Jaeger et al., 2014, Candemir et al., 2014], OCTA-500 [Li et al., 2019], CamVid [Brostow et al., 2009]) using large pre-trained models. We evaluate performance on a subset of 200 test images across varying perturbation dimensions and magnitudes, with results shown in Figure 2. Model details and probabilistic guarantees are summarized in Table 2, where we show the following $\langle \epsilon, \ell, m \rangle$ guarantee:

$$\Pr\left[\Pr\left[\mathsf{P}\right] \geq \delta_1\right] \geq \delta_2, \quad \text{where } \mathsf{P} := \text{"} \left\{ \begin{array}{l} \text{Given an image } x, \text{ the perturbation magnitude and dimension } e, r, \\ \text{the computed robustness value } \mathbf{RV} \text{ from our technique is valid.} \end{array} \right\} \text{"}$$

We also present a demo for the status of all pixels in the segmentation mask in Figures 4,5 and 6.

To assess the conservatism, we examine the projection bounds $[\underline{y}, \overline{y}]$ introduced in Algorithm 2. Specifically, we sample $10^6$ adversarial examples from $x^{\text{adv}} \overset{\mathcal{W}}{\sim} \mathbf{I}$ and use them to report:

- Empirical miscoverage $\hat{\epsilon}$, that is the percentage of events, where $f(\mathbf{vec}(x^{\text{adv}})) \notin [\underline{y}, \overline{y}]$.

- Empirical bound $[\hat{y}, \overline{\hat{y}}]$, via component-wise minima/maxima of $f(\mathbf{vec}(x^{\text{adv}}))$ across samples.

The degree of conservatism can be assessed by comparing $[\hat{y}, \overline{\hat{y}}]$ with $[\underline{y}, \overline{y}]$. To visualize it, we plot,

$$\mathsf{bound\_ratio}(k) = (\overline{\hat{y}}(k) - \hat{y}(k))/(\overline{y}(k) - \underline{y}(k)), \quad k = 1, \ldots, h \times w \times L$$

for logit components in Figure 8 (Appendix E) as histogram. Since $10^6$ inferences is costly, we study the conservatism on three cases from Figure 2, spanning our smallest to largest configurations.

---

[6]Compared to our approach, techniques based on randomized smoothing have the advantage of providing guarantees without assuming any prior distribution. The downside, however, is that their guarantees are derived for an approximate version of the model that implicitly assumes a prior Gaussian distribution (see Appendix D).

**Algorithm 2:** Detection of the pixel status

**Input: I**, $x$, $f$, $\mathcal{W}$, $\epsilon$, Hyper-parameters$(\ell, m)$
**Output:** The status of pixels

// Project the Starset
$[\underline{y}, \overline{y}] \leftarrow \mathcal{R}_f^{\epsilon}(\mathcal{W}; \ell, m)$

**foreach** $(i, j) \in \{1{:}h\} \times \{1{:}w\}$ **do**
    $l^* \leftarrow \arg\max_l \underline{y}(i, j, l)$
    **if** $\underline{y}(i, j, l^*) \leq \max_{l \neq l^*} \overline{y}(i, j, l)$ **then**
        $\lfloor$ Label pixel $(i, j)$ as unknown
    **else if** $l^* = \mathsf{SSN}(x)(i, j)$ **then**
        $\lfloor$ Label pixel $(i, j)$ as robust
    **else**
        $\lfloor$ Label pixel $(i, j)$ as non-robust

**return** *Pixel-wise labeling*

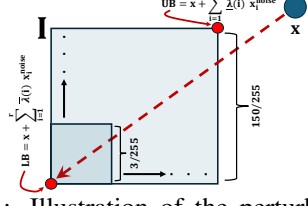

Figure 1: Illustration of the perturbation set **I** associated with an $r$-dimensional darkening adversary applied to image $x$. The set is constructed by applying independent perturbations to all $nc$ channels across $r'$ selected pixels ($r = nc \times r'$), each having R, G, and B intensities above $150/255$. The lower bound **LB** corresponds to the maximum darkening case (channel intensities set to zero), whereas the upper bound **UB** represents the minimum darkening limit.

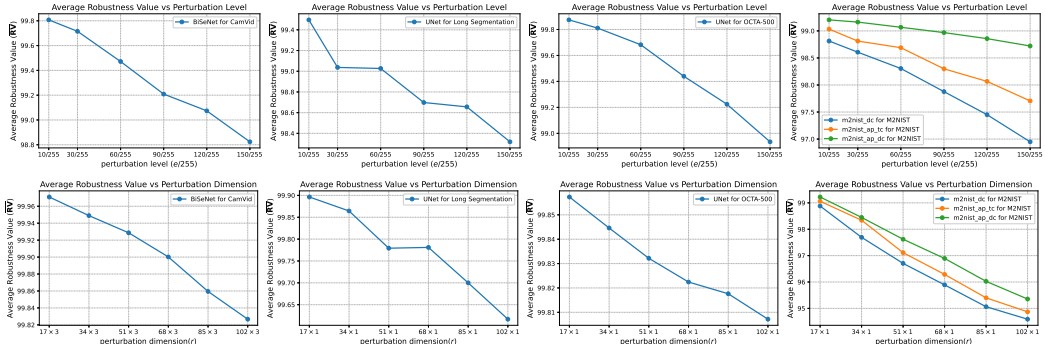

Figure 2: Shows $\overline{\mathbf{RV}}$ versus perturbation level $e$ (top row) and perturbation dimension $r$ (bottom row). In the top row, $r = 102 \times 3$ for CamVid, $102 \times 1$ for Lung Segmentation and OCTA-500, and $17 \times 1$ for M2NIST experiment. In the bottom row, $e$ is fixed at $3/255$ across all experiments. The average robustness value $\overline{\mathbf{RV}}$ is averaged over 200 test images.

## 5 Limitations, Future Works & Conclusion

**Limitation**. The **first** limitation of our technique is the assumption of a prior distribution $\mathcal{W}$, which may not hold in all deployment scenarios. However, this issue can be addressed at a low cost by replacing CI with robust-CI [Cauchois et al., 2024], which preserves our guarantees even when the deployment distribution shifts within a known $f$-divergence ball [Cauchois et al., 2024]. The **second** limitation of our technique is the need to recompute the calibration set for each test point, which accounts for the majority of the computation time in our approach. **Finally**, while replacing the naive technique with the surrogate-based technique improves accuracy, it also introduces some drawbacks. It increases runtime, as it requires performing PCA, training, and deterministic reachability analysis, and it also limits scalability with respect to input dimensionality, since higher-dimensional settings make both training and deterministic reachability more challenging. Therefore, in cases where the perturbation dimension is very large, we fall back to the naïve technique for scalability.

**Future Work**. The main advantage of the surrogate-based technique is that its guarantees also capture the interrelations among mask pixels, offering the potential to verify **spatial specifications** over SSNs—an aspect not emphasized in this work. In addition, incorporating robust-CI can address the issue of **distribution shift** in deployment settings. Therefore, we consider and explore both of these mentioned extensions in our future research on this topic.

**Conclusion**. We do not provide deterministic guarantees, and our reachable set remains susceptible to violations; however, it offers provable probabilistic assurances, as evidenced in Appendix F. Despite this limitation, our approach is both scalable and data-efficient, delivering meaningful safety assurances in regions where existing deterministic methods are computationally infeasible. We demonstrated these capabilities through a series of numerical experiments presented in this paper.

# 6 Acknowledgement

This material is based upon work supported by internal funding from Vanderbilt University for medical imaging research and the National Science Foundation (NSF) under Award Numbers FMitF-2220401, 2220426, and 2443803. Any opinions, findings, and conclusions or recommendations expressed in this paper are those of the authors and do not necessarily reflect the views of Vanderbilt or NSF.

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

## A   Double-Step Probabilistic Guarantee

The necessary steps to establish the two-sided probabilistic guarantee, along with the flexibility in choosing the dataset size $m$, rank $\ell$, and failure probability $\epsilon$, begin with the theory of order statistics [David and Nagaraja, 2004]. This flexibility means that the double-step probability constraint remains valid regardless of the particular values chosen for $m$, $\ell \leq m$, and $\epsilon \in [0,1]$. The following lemma introduces the concept of order statistics.

**Lemma A.1 (From David and Nagaraja [2004])** *Consider $m$ independent and identically distributed (i.i.d.), real-valued data points $x \in [0,1]$ drawn from uniform distribution $x \sim \mathcal{U}$. Suppose we sort them in ascending order and denote the $\ell^{th}$ smallest number by $x_\ell$,(i.e., we have $x_1 < x_2 < \ldots < x_m$). Let $\mathbf{Beta}(\alpha, \beta)$ denote the Beta distribution[7]. Then the uniform random variable at rank $\ell$ follows:*

$$x_\ell \sim \mathbf{Beta}(\ell, m + 1 - \ell)$$

In the context of conformal inference Lemma A.1, has been extended from a uniform and domain-bounded distribution to a general distribution as follows:

**Lemma A.2 (From Vovk et al. [2005])** *Consider $m$ independent and identically distributed (i.i.d.), real-valued data points $R$ drawn from some distribution $R \sim \mathcal{D}$. Suppose we sort them in ascending order and denote the $\ell^{th}$ smallest number by $R_\ell$, (i.e., we have $R_1 < R_2 < \ldots < R_m$). Let $\mathbf{Beta}(\alpha, \beta)$ denote the Beta distribution. Then, for an arbitrary $R_{m+1}$ drawn from the same distribution $\mathcal{D}$, the following holds:*

$$\Pr\left[R_{m+1} < R_\ell\right] \sim \mathbf{Beta}(\ell, m + 1 - \ell), \quad 1 \leq \ell \leq m. \tag{8}$$

This result is motivated by two key observations: (1) for a continuous distribution $R \sim \mathcal{D}$, the cumulative probability $\Pr[R \leq r]$ is uniformly distributed on $[0,1]$, and (2) sorting the samples $R_1 < R_2 < \ldots < R_m$ implies that the corresponding cumulative probabilities $\Pr[R \leq R_\ell]$ are also ordered, i.e., $\Pr[R \leq R_1] < \Pr[R \leq R_2] < \ldots < \Pr[R \leq R_m]$. These observations implies that for all $\ell = 0, 1, 2, \ldots, m$, the random variable $\delta := \Pr[R \leq R_\ell]$ satisfies the conditions outlined in Lemma A.1.

In simple terms, this means that while a fixed dataset $R_1, R_2, \ldots, R_m$ is not sufficient to determine the exact value of $\delta = \Pr[R < R_\ell]$, we can infer that the value of $\delta$ must follow a Beta distribution that is solely defined based on $m$ and $\ell$. This allows us to tune the parameters $m$ and $\ell$ to provide strong guarantees for the coverage $R < R_\ell$ where $R_\ell$ is obtained from our fixed sampled dataset.

This guarantee can be equivalently expressed as a two-sided probabilistic bound. Specifically, let $\mathsf{betacdf}_\delta(\ell, m + 1 - \ell)$ denote the cumulative distribution function (CDF) of the Beta distribution—also known as the regularized incomplete beta function—evaluated at $\delta$. Then, based on the definition of cumulative density function, CDF, for any confidence $\delta_1 \in [0,1]$ and any arbitrary distribution $R \sim \mathcal{D}$, the following inequality holds:

$$\Pr\left[\ \Pr[\ R \leq R_\ell\ ] > \delta_1\ \right] > 1 - \mathsf{betacdf}_{\delta_1}(\ell, m + 1 - \ell).$$

Note that the key assumption is that all variables $R, R_1, R_2, \ldots, R_m$ are independently drawn from the same distribution $\mathcal{D}$, and that the previously sampled values are sorted such that $R_1 < R_2 < \ldots < R_m$. Importantly, the resulting probabilistic guarantee remains valid for any arbitrary choice of sample size $m$, rank $\ell$, and confidence level $\delta_1$, provided they lie within appropriate bounds.

## B   A Clarifying Example for Conformal Inference

Assume an arbitrary distribution $R \sim \mathcal{D}$. We sample a calibration dataset of size $m = 100,000$, we sort the samples as $R_1 < R_2 < \ldots < R_{100,000}$ and we select the sample at rank $\ell = 99,999$. In

---

[7]The Beta distribution is a family of continuous probability distributions defined on the interval $0 \leq x \leq 1$ with shape parameters $\alpha$ and $\beta$, and with probability density function $f(x; \alpha, \beta) = \frac{x^{\alpha-1}(1-x)^{\beta-1}}{B(\alpha, \beta)}$, where the constant $B(\alpha, \beta) = \frac{\Gamma(\alpha)\Gamma(\beta)}{\Gamma(\alpha+\beta)}$ and $\Gamma(z) = \int_0^\infty t^{z-1} e^{-t} \mathbf{d}t$ is the Gamma function.

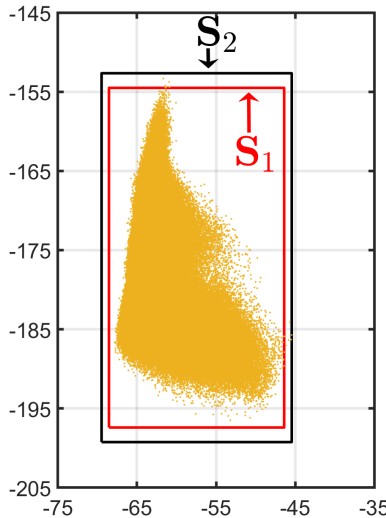

Figure 3: Shows two Reachsets with different $\langle \epsilon, \ell, m \rangle$ guarantees.

this case, given an unseen draw $R^{\text{unseen}}$ from $\mathcal{D}$ for a miscoverage level $\epsilon = 0.0001$, the $\langle \epsilon, \ell, m \rangle$ guarantee for logical statement $P(\ell, m) = [R^{\text{unseen}} < R_\ell]$ is:

$$\Pr[\ \Pr[\ R^{\text{unseen}} < R_{99,999}\ ]\ >\ 0.9999\ ]\ >\ 0.9995008$$

which is valid for any calibration dataset $\mathbf{M} = \{R_1, R_2, \ldots, R_{100,000}\}$ sampled from distribution $\mathcal{D}$ if and only if $R^{\text{unseen}}$ is also sampled from $\mathcal{D}$, and $R_{99,999}$ is the $99,999^{\text{th}}$ smallest member of $\mathbf{M}$.

## C   A Clarifying Example on Naive Approach

Since the naive technique is architecture-agnostic we apply that for the reachability analysis of a feedforward ReLU neural network with 60 hidden-layers of width 100, an input layer of dimension 784 and an output layer of dimension 2. The set of inputs is $\mathbf{I} := [0, 1]^{784}$ and we generate both calibration and train datasets by sampling $\mathbf{I}$ uniformly. e.g., $\mathcal{W}, \mathcal{W}'$ are both uniform distributions.

**1:** We firstly generate a calibration dataset of size $m = 200,000$ and a train dataset of size $t = 10,000$, we also consider the rank $\ell = 199,998$ and we target the miscoverage level of $\epsilon = 0.0001$. In this case we compute a reachable set, $\mathbf{S}_1$ with the following $\langle \epsilon, \ell, m \rangle$ guarantee.

$$x \overset{\mathcal{W}}{\sim} \mathbf{I} \ \Rightarrow\ \Pr[\ \Pr[\ f(x) \in \mathbf{S}_1\ ] > 0.9999\ ] > 0.9999995.$$

**2:** Secondly, we set $m = 8464287$, $\ell = 8464286$ and $\epsilon = 0.000002$ which results in another reachset $\mathbf{S}_2$ with the following $\langle \epsilon, \ell, m \rangle$ guarantee.

$$x \overset{\mathcal{W}}{\sim} \mathbf{I} \ \Rightarrow\ \Pr[\ \Pr[\ f(x) \in \mathbf{S}_2\ ] > 0.999998\ ] > 0.9999992.$$

Figure 3 shows the reachable sets $\mathbf{S}_1$ and $\mathbf{S}_2$ in the presence of $10^6$ new simulations, $f(x), x \overset{\mathcal{W}}{\sim} \mathbf{I}$, for validation. Our calculations show that the proportion of points lying outside of $\mathbf{S}_1$ and $\mathbf{S}_2$ are 0.000013 and 0.000001, respectively, which aligns well with the proposed $\langle \epsilon, \ell, m \rangle$ guarantees.

## D   Additional material for RQ1

In this section, we present an overview of our conformal inference–based technique and contrast it with methods based on randomized smoothing, using a unified terminology to make the comparison clearer for readers.

**Overview 1**. In Fischer et al. [2021], the authors introduced a probabilistic method to verify Semantic Segmentation Neural Networks. Inspired by Cohen et al. [2019], they introduced randomness via a

Gaussian noise $\nu \sim \mathcal{N}(0, \sigma^2) \in \mathbb{R}^{h \times w \times nc}$, applied to the input, and constructed a smoothed version of the segmentation model, denoted $\overline{\mathsf{SSN}}(x)(i,j)$, for each mask pixel $(i,j)$:

$$\overline{\mathsf{SSN}}(x)(i,j) = c_A(i,j) = \arg\max_{c \in \mathbf{L}} \Pr_{\nu \sim \mathcal{N}(0,\sigma^2)} [\mathsf{SSN}(x+\nu)(i,j) = c]$$

They then established that, with confidence level $\delta_2$, the smoothed prediction $\overline{\mathsf{SSN}}(x)(i,j)$ is robust within an $\ell_2$ ball $\mathbf{B}_{\bar{r}_{i,j}}(x)$ of radius $\bar{r}_{i,j} = \sigma\Phi^{-1}(\underline{p_A}(i,j))$, where $\underline{p_A}(i,j)$ is a lower bound on the class probability: $p_A(i,j) = \Pr_{\nu \sim \mathcal{N}(0,\sigma^2)} [\mathsf{SSN}(x+\nu)(i,j) = c_A(i,j)]$. Here, $\Phi^{-1}(.)$ is the inverse CDF of normal distribution. The corresponding probabilistic guarantee becomes:

$$\forall x' \in \mathbf{B}_{\bar{r}_{i,j}}(x) : \Pr\left[\overline{\mathsf{SSN}}(x')(i,j) = c_A(i,j)\right] \geq \delta_2$$

However, certifying all mask pixels simultaneously is challenging. To enable global guarantees, a more conservative definition of smoothing is adopted: a fixed threshold $\kappa \in [0.5, 1]$ is used for all mask locations $(i,j)$ which implies:

$$\overline{\mathsf{SSN}}(x)(i,j) = c_A(i,j), \quad \text{where} \quad \Pr_{\nu \sim \mathcal{N}(0,\sigma^2)} [\mathsf{SSN}(x+\nu)(i,j) = c_A(i,j)] \geq \kappa$$

This leads to a unified radius $r = \sigma\Phi^{-1}(\kappa)$ for the input space. While this simplifies the formulation, it introduces the concept of **abstained** or uncertifiable pixels—those that cannot meet the desired confidence threshold. To mitigate the issue of multiple comparisons (i.e., union bounds), the authors propose statistical corrections such as the Bonferroni and Holm–Bonferroni methods [CE, 1936, Holm, 1979]. Assuming the set of certifiable pixels is defined as $\mathbf{CERT} = \{(i,j) \mid (i,j) \text{ is certifiable}\}$, they propose the final guarantee in the following form:

$$\forall x' \in \mathbf{B}_r(x) : \Pr\left[\bigwedge_{(i,j) \in \mathbf{CERT}} \mathsf{P}(i,j)\right] \geq \delta_2, \quad \mathsf{P}(i,j) := \left\{"\overline{\mathsf{SSN}}(x')(i,j) = c_A(i,j)"\right\}$$

This method is referred to as SEGCERTIFY. A key limitation of this technique is the presence of abstained pixels. The authors in Anani et al. [2024] addressed this by proposing an adaptive approach, ADAPTIVECERTIFY, which reduces the number of uncertifiable pixels. However, the guarantees proposed in Fiacchini and Alamo [2021], Anani et al. [2024] apply only to the smoothed model, not the base model. To express this in terms of the base model, we define the following problem:

*Problem 1*. For a given image $x$, noise $\nu \sim \mathcal{N}(0, \sigma^2)$, and a threshold $\kappa \in [0.5, 1]$, define $c_A(i,j) \in \mathbf{L}$ such that: $\Pr_{\nu \sim \mathcal{N}(0,\sigma^2)} [\mathsf{SSN}(x+\nu)(i,j) = c_A(i,j)] \geq \kappa$. Then for any $x' \in \mathbf{B}_r(x)$ with $r = \sigma\Phi^{-1}(\kappa)$ and confidence level $\delta_2$, we want to show the following guarantee, where $\mathbf{CERT}$ will be also determined through the verification process:

$$\Pr\left[\bigwedge_{(i,j) \in \mathbf{CERT}} \Pr_{\nu \sim \mathcal{N}(0,\sigma^2)} [\mathsf{SSN}(x'+\nu)(i,j) = c_A(i,j)] \geq \kappa\right] \geq \delta_2$$

**Overview 2**. In our approach, for a given image $x$ and input set $\mathbf{B}_r(x)$, we perform a probabilistic reachability analysis with a $\langle\epsilon, \ell, m\rangle$ guarantee. Due to the presence of conservatism in our reachability technique, some mask pixels are marked as **robust** (i.e., certifiable), while others cover multiple classes and are considered **unknown** or uncertifiable. Let $\mathbf{CONFORMAL\_CERT} = \{(i,j) \mid (i,j) \text{ is certifiable}\}$. Then our verification objective is:

*Problem 2*. Given an image $x$, input set $\mathbf{B}_r(x)$, and sampling distribution $x' \overset{\mathcal{W}}{\sim} \mathbf{B}_r(x)$, for coverage level $\delta_1 = 1 - \epsilon \in [0, 1]$, hyper-parameters $\ell, m$ and confidence level $\delta_2 = 1 - \mathrm{betacdf}_{\delta_1}(\ell, m+1-\ell)$, we aim to show the following $\langle\epsilon, \ell, m\rangle$ guarantee, where $\mathbf{CONFORMAL\_CERT}$ will be also determined through the verification process:

$$\Pr\left[\Pr\left[\bigwedge_{(i,j) \in \mathbf{CONFORMAL\_CERT}} \mathsf{SSN}(x')(i,j) = \mathsf{SSN}(x)(i,j)\right] \geq \delta_1\right] \geq \delta_2$$

Both guarantee formulations have their respective advantages and limitations. An advantage of our guarantee formulation is that it is defined directly on the base model $\mathsf{SSN}(x')(i,j)$ while randomized smoothing considers $\mathsf{SSN}(x'+\nu)(i,j)$. To make the randomized smoothing formulation operate directly on the base model, one must set $\nu = 0$, which implies $\sigma = 0$ and consequently $r = \sigma\Phi^{-1} = 0$. As a result, the perturbation ball $\mathbf{B}_r(x)$ collapses to a singleton, leaving no space for verification. On the other hand, the advantage of randomized smoothing lies in its consideration of the worst-case distribution $x' \in \mathbf{B}_r(x)$, whereas our methodology assumes a prior distribution $x' \overset{\mathcal{W}}{\sim} \mathbf{B}_r(x)$.

# E   Additional material for RQ2

Here, we present the additional graphs and visualizations referenced in RQ2.

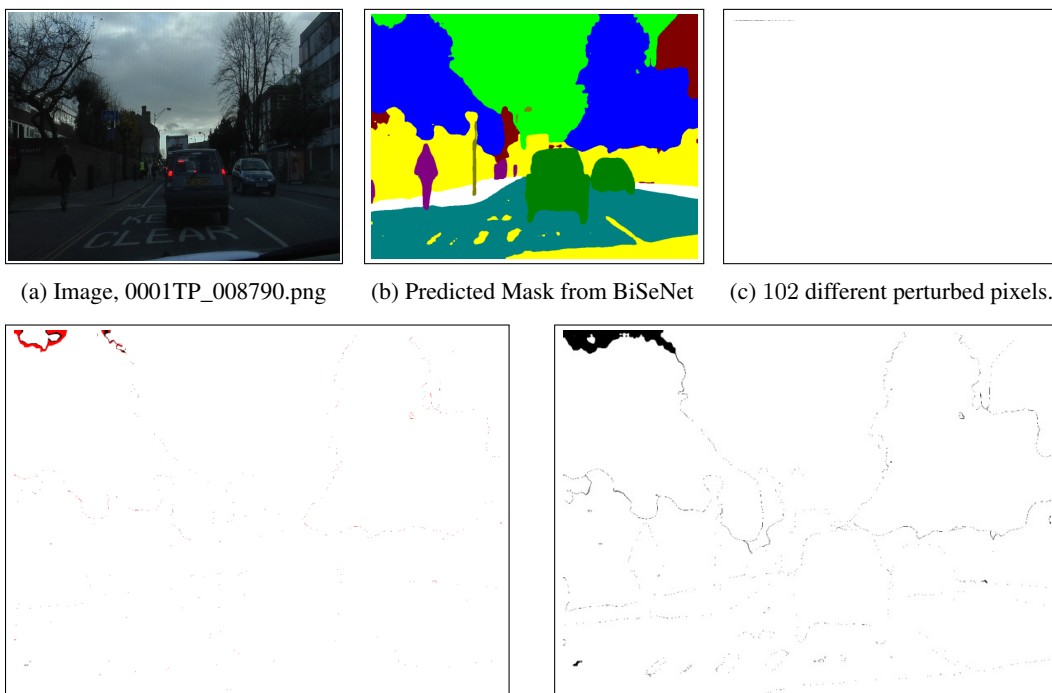

(a) Image, 0001TP_008790.png    (b) Predicted Mask from BiSeNet    (c) 102 different perturbed pixels.

(d) Unknown and non-robust pixels for $e = 10/255$.    (e) Unknown and non-robust pixels for $e = 150/255$.

Figure 4: Visualization for verification on BiSeNet for a specific test image from the CamVid dataset. (a) The test image used for verification. (b) The segmentation mask predicted by BiSeNet for this image. (c) The pixels selected for perturbation (in black) on the test image (We sampled 102 pixels where the R, G, and B intensities were all above $150/255$, forming a perturbation set $\mathbf{I} \subset \mathbb{R}^{102\times3}$). (d,e) Display the robust (white), non-robust (red), and unknown (black) pixels for the perturbation set $\mathbf{I}$ as described in Figure 1, with perturbation magnitudes $e = 10/255$ and $e = 150/255$, respectively. The figure illustrates that the model is not sufficiently robust on the borders between different classes.

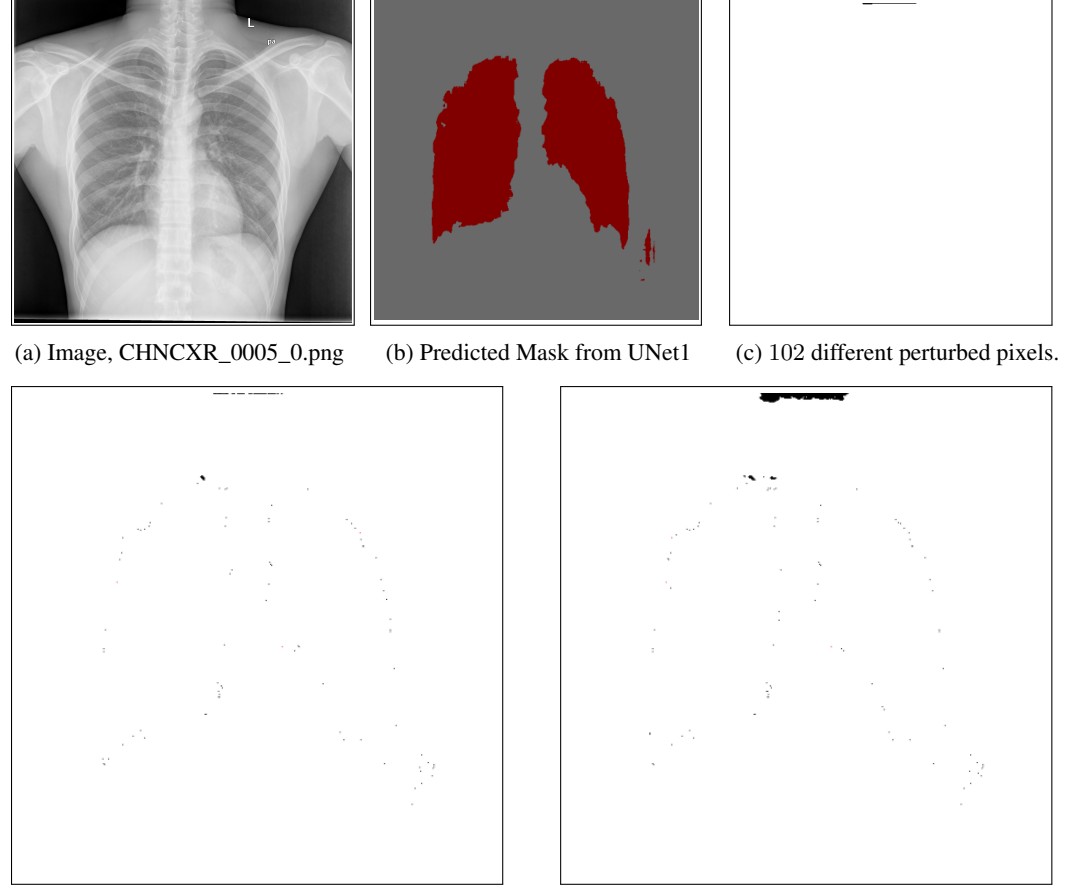

(a) Image, CHNCXR_0005_0.png  (b) Predicted Mask from UNet1  (c) 102 different perturbed pixels.

(d) Unknown and non-robust pixels for $e = 10/255$.  (e) Unknown and non-robust pixels for $e = 150/255$.

Figure 5: Visualization for verification on UNet1 for a specific test image from the Lung Segmentation dataset. (a) The test image used for verification. (b) The segmentation mask predicted by UNet1 for this image. (c) The pixels selected for perturbation (in black) on the test image (We sampled 102 pixels where the Gray intensities were above $150/255$, forming a perturbation set $\mathbf{I} \subset \mathbb{R}^{102 \times 1}$). (d,e) Display the robust (white), non-robust (red), and unknown (black) pixels for the perturbation set $\mathbf{I}$ as described in Figure 1, with perturbation magnitudes $e = 10/255$ and $e = 150/255$, respectively.

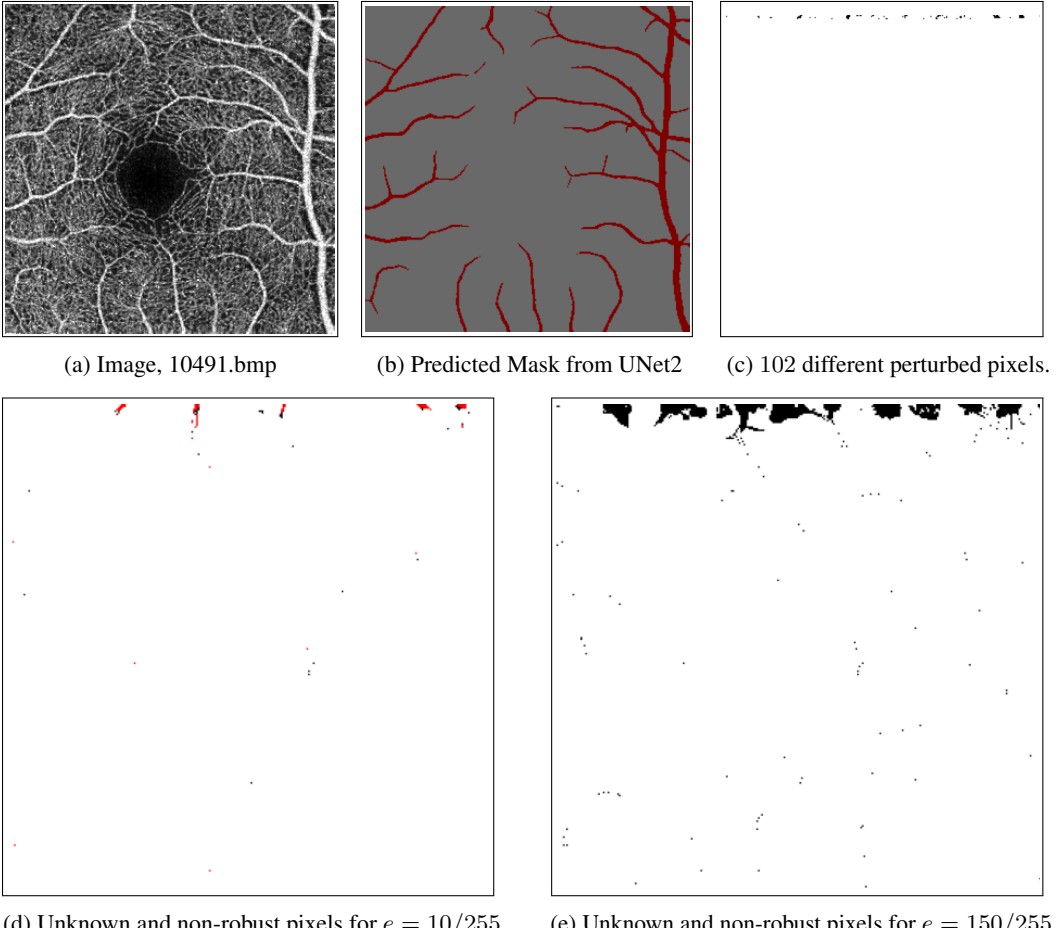

(a) Image, 10491.bmp     (b) Predicted Mask from UNet2     (c) 102 different perturbed pixels.

(d) Unknown and non-robust pixels for $e = 10/255$.     (e) Unknown and non-robust pixels for $e = 150/255$.

Figure 6: Visualization for verification on UNet2 for a specific test image from the OCTA-500 dataset. (a) The test image used for verification. (b) The segmentation mask predicted by UNet2 for this image. (c) The pixels selected for perturbation (in black) on the test image (We sampled 102 pixels where the Gray intensities were above $150/255$, forming a perturbation set $\mathbf{I} \subset \mathbb{R}^{102 \times 1}$). (d,e) Display the robust (white), non-robust (red), and unknown (black) pixels for the perturbation set $\mathbf{I}$ as described in Figure 1, with perturbation magnitudes $e = 10/255$ and $e = 150/255$, respectively.

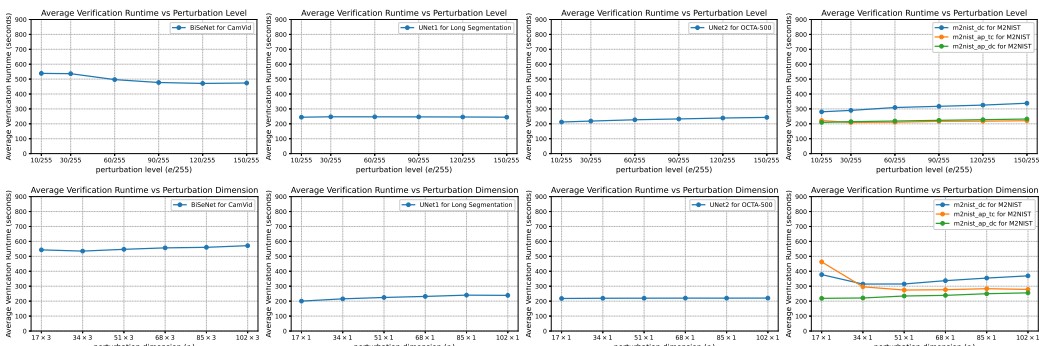

Figure 7: Shows the average run time versus perturbation level $e$ (top row) and perturbation dimension $r$ (bottom row). In the top row, $r = 102 \times 3$ for CamVid, $102 \times 1$ for Lung Segmentation and OCTA-500, and $17 \times 1$ for M2NIST experiment. In the bottom row, $e$ is fixed at $3/255$ across all experiments. The runtimes are averaged over 200 test images in each experiment. This graph shows our verification runtime is only slightly sensitive the perturbation magnitude and direction $e, r$.

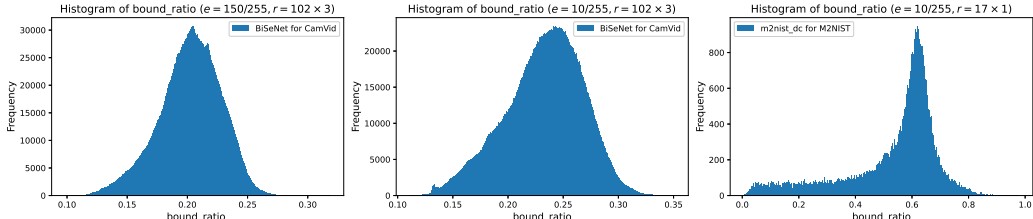

Figure 8: Histogram of $n = h \times w \times L$ numbers (called bound_ratio) shown to present conservatism analysis for our technique. These numbers are the ratio between the length of the empirical bounds and the length of our proposed bounds, bound_ratio$(k) = (\overline{\hat{y}}(k) - \hat{y}(k))/(\overline{y}(k) - \underline{y}(k))$ where $k = 1, \ldots, n$. The empirical bounds $[\hat{y}(k), \overline{\hat{y}}(k)]$ are estimated using $10^6$ random samples. We conduct conservatism analysis across three experiments from Figure 2: one experiment with the m2nist_dc model trained on M2NIST dataset where the experiment considers perturbation magnitude $e = 10/255$, and dimension $r = 17 \times 1$; and two experiments with the BiSeNet model trained on CamVid dataset where the experiment considers perturbation magnitue $e = 10/255 \ \& \ 150/255$ and dimension $r = 102 \times 3$. These cases span the smallest to the largest configurations we addressed in our case studies in Figure 2. The empirical miscoverage $\hat{\epsilon}$ was $\hat{\epsilon} = 7 \times 10^{-6}$ for M2NIST experiment and $\hat{\epsilon} = 0$ for both experiments on CamVid dataset. This shows in all cases $\hat{\epsilon} < \epsilon$ where $\epsilon = 10^{-4}$ for M2NIST and $\epsilon = 10^{-3}$ for CamVid, as outlined in Table 2. In this histogram, we omit components $k$ whose normalization factors $\tau_k$ are smaller than $\tau^*$, since their magnitudes are negligibly small compared to the other components, and their conservatism has little effect on the volume difference between the sets.

## F   Looking at Violating Samples Obtained in Our Experiments

Our probabilistic guarantees inherently imply that a small probability of violation exists; however, the key contribution of our approach is that it provides formal guarantees on the likelihood of such violations. One example of these events appears in Appendix C, where the red and black boundaries are violated by 13 and 1 output samples, respectively, out of a total of $10^6$ generated outputs. Another instance is shown in Figure 8, where 7 out of $10^6$ sampled outputs violate the reachable set computed for the M2NIST experiment. Importantly, all of these violations remain consistent with the probabilistic bounds established by our theoretical guarantees.

## G   Additional Research Questions

**RQ3 ($2^{nd}$ Abblation Study): How Do Surrogate Models Affect the Level of Conservatism in our method?**   To investigate this, we conducted a classification-based verification task using a model submitted to VNN-COMP 2024 [Brix et al., 2024] by the $\alpha$-$\beta$-CROWN team, trained on the CIFAR-100 dataset. We refer to this model as ResNet_large, which contains 8 residual blocks, 19 convolutional layers, and 2 linear layers. We sample a calibration dataset of size $m = 100,000$, set the rank to $\ell = 99,999$, and choose a miscoverage level $\epsilon = 10^{-4}$, which corresponds to a coverage level of $\delta_1 = 1 - \epsilon = 0.9999$. This setup yields a confidence level of $\delta_2 = 0.9995$. In this experiment, given an image $x \in \mathbb{R}^{32 \times 32 \times 3}$ with label $\ell \in 1, \ldots, 100$—corresponding to a logit vector $f(\mathbf{vec}(x)) \in \mathbb{R}^{100}$—we aim to verify the following specification:

$$\Pr\left[\Pr\left[\mathsf{P}\right] \geq \delta_1\right] \geq \delta_2, \quad \text{where } \mathsf{P} := \left\{ \begin{matrix} \text{Given an image } x \text{ perturbed within an } \ell_\infty \text{ ball of radius } e/255 \text{ and label } \ell, \\ \text{the predicted label remains unchanged for all perturbations within the ball} \end{matrix} \right\}$$

Our reachability-based verification method inherently introduces some conservatism. In this experiment, we investigate how incorporating surrogate models—specifically, a ReLU-based surrogate—can reduce that conservatism compared to a naive approach. Lower conservatism enables verification under larger perturbation magnitudes $e/255$. To evaluate this, we vary $e$ across $1/255$ to $30/255$ and test both methods on 200 sampled images. The naive method fails to verify beyond certain perturbation levels, while the surrogate-based approach continues to succeed under larger perturbations, achieving the same probabilistic guarantee. Thus given a unique perturbation magnitude $e$ we run the verification on 200 events (200 images) and count the percentage of successful verification for both methods. Figure 9 presents this percentage of successful verifications for each technique across

perturbation levels, demonstrating that the ReLU surrogate substantially improves the method's applicability by reducing the level of conservatism. As previously mentioned, our verification runtimes vary minimally across different images and perturbation levels, so we report average values. The naive method took an average of 6 seconds per verification, while the ReLU surrogate method required 44 seconds.

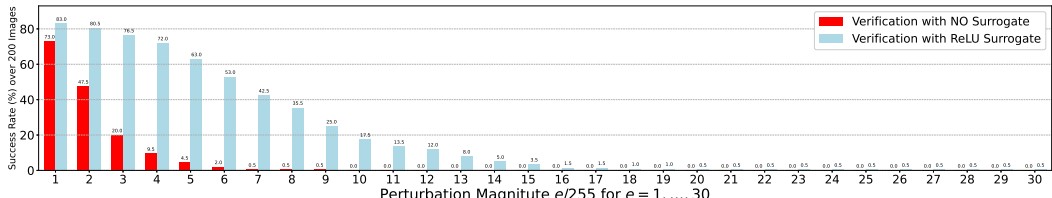

Figure 9: This figure compares the conservatism of verification with and without the ReLU surrogate. We evaluated classification robustness over perturbations bounded by $\ell_\infty$ balls with magnitudes $e = 1/255, 2/255, \ldots, 30/255$, across 200 images. Each block represents the percentage of cases where the verified label remained unchanged under perturbation. The results show that using a ReLU surrogate significantly improves verification at larger perturbation levels, due to reduced conservatism in the reachability analysis.

**RQ4: Do State-of-the-Art Deterministic Verification Techniques Scale to the Level Achieved by Our Method on Complex and High-Dimensional SSN Models?**  We applied techniques such as $\alpha$-$\beta$-CROWN [Zhou et al., 2024] and NNV [Tran et al., 2021] to our segmentation experiments using the UNet1, UNet2, and BiSeNet models in RQ1. However, due to the size of these models and the high perturbation levels considered, both methods encountered out-of-memory errors and failed to complete verification on the same hardware used for our approach. This highlights that while deterministic guarantees are often preferable, they may not always be computationally practical. In such cases, our probabilistic verification method for SSNs offers a scalable and effective alternative.

