# OpenReview forum: "Scaling Data-Driven Probabilistic Robustness Analysis for Semantic Segmentation Neural Networks"
_NeurIPS.cc/2025/Conference — NeurIPS 2025 poster_

### Official Review · Reviewer_BgoC · 2025-06-25

**Clarity:** 2
**Significance:** 3
**Originality:** 3
**Rating:** 4
**Confidence:** 2

**Summary:**

The paper proposes a three-stage pipeline that delivers pixel-level robustness certificates with finite-sample calibration for full-resolution semantic segmentation networks (SSNs). First, in the naïve conformal reachability stage, a calibration set of size m is used to construct an axis-aligned hyper-rectangle that, with user-defined risk level ϵ, contains the SSN’s logits under an input-uncertainty distribution. Second, in the surrogate + star-set inflation stage, a small ReLU surrogate model g is trained; its exact star-set reach set is computed using NNV and then expanded via a Minkowski sum with an error-bound hyper-rectangle derived through conformal inference, resulting in a significantly tighter reach set. Third, deflation-based PCA is used to address the extremely high dimensionality of SSN outputs: the top-N principal directions are learned, allowing g to operate in a low-dimensional latent space and be lifted back linearly to the original space. The final reach set is projected onto per-pixel logits to determine whether each pixel is robust, non-robust, or uncertain. Experiments on M2NIST, Lung_Segmentation, CamVid, and a CIFAR-100 classifier show that the method certifies ≥99.9% robustness value (RV) at ≥0.999 confidence, with runtimes ranging from 3 seconds to 60 minutes on a single 48 GB GPU.

**Questions:**

Hyper-parameter sensitivity: Please sweep calibration size *m* and PCA rank *N*; plot RV vs. runtime.  How fast does the guarantee degrade if *m* is cut 10×? How sensitive is the model to m and N?

Dataset coverage: Would the model also work on Cityscapes and ADE20K since those are important datasets in the field?

**Ethical Concerns:**

["NO or VERY MINOR ethics concerns only"]

**Final Justification:**

After considering the authors' rebuttal and the discussions, I maintain my original evaluation. Here is a summary of the issues that were resolved and of those that remain:

The authors clarified that Algorithm 1 includes two principled stopping criteria in practice—variance threshold and a hard cap on the number of directions. This resolves my concern about clarity in algorithmic description and adds confidence in the implementation robustness.

The authors committed to adding an ablation study on hyperparameters such as calibration size m and PCA rank N. Even though not present in the original submission, this plan addresses the concern.

The authors provided a new evaluation on the Cityscapes dataset. This strengthens the significance and applicability of the method.

However, The core techniques—CI, PCA, and surrogate modeling—are not new. The paper’s value lies in their integration and adaptation to the challenging SSN setting. While effective, this limits the theoretical novelty.

My Recommended Score: 4 (Borderline Accept)

**Limitations:**

yes

**Quality:**

2

**Strengths And Weaknesses:**

Quality:
+ S: Runtime scales to 960 × 720 outputs where deterministic tools have difficulties.
+ S: Empirical study uses three metrics (RV, RS, RIoU) and reports per-experiment (ℓ, *m*) choices (Table 3).
- W: The choice of the number of PCA components N is treated as a fixed hyperparameter without a principled selection method, which may affect generality and reproducibility. No ablation on PCA rank N and hyper-parameter sensitivity is unknown.

Clarity:
+ S: Incremental step by step explanation naïve - surrogate - PCA.
- W: Minor issues—Algorithm 1 lacks a stop-criterion.

Significance:
+ S: The method significantly reduces runtime compared to pixel-wise conformal prediction, while scaling to high-dimensional outputs using PCA and star sets
- W: Evaluation omits standard datasets Cityscapes/ADE20K used in prior CP work  and smoothing papers. At least they should appear in the discussion regarding their applicability due to their importance.

Originality:
+ S: Novel integration of CI with star-set reachability via combining these techniques for segmentation.
+ S: PCA-deflation surrogate training adapts sparse-PCA ideas to verification.
- W: The individual ingredients (CI, star sets, PCA) exist; novelty stems from synergy more than new theory.

---

> ### Author Rebuttal · Authors · 2025-07-31
>
> We appreciate your detailed feedback
>
> ---
> ---
>
> ***Response to Reviewer’s Comments on Weaknesses***
>
> **W (Quality):**
> Algorithm 1 is presented in a simplified form for clarity, but in our implementation, the SGD-based deflationary PCA procedure includes two termination criteria:
>
> 1. The first criterion is based on variance: the algorithm stops when the optimal variance captured in the current iteration falls below 1% of the optimal variance recorded in the first iteration. Since each iteration involves an optimization with no local maxima, it is guaranteed that the optimal variance captured in each subsequent iteration decreases monotonically.
>
> 2. The second criterion is a fixed upper bound on the number of principal directions, which we set to $N = 1000$. If the number of extracted directions exceeds this value, the algorithm terminates automatically.
>
> The PCA process stops as soon as **either** of these conditions is satisfied.
>
> We agree that a principled analysis of sensitivity to the PCA rank $N$ is valuable. Therefore, we will include an ablation study in the appendix to evaluate the effect of this and other hyperparameters (e.g., $m$, $N$) on the performance and robustness of our method.
>
> Regarding the choice of $N = 1000$: In case the correlation matrix of logit dimensions is not approximately sparse, this number may sometimes be insufficient for achieving an accurate surrogate model. However, this does **not** pose a significant problem for our framework. The goal of the surrogate model is **not** to perfectly approximate the SSN but rather to **reduce the conservatism** that exists in the naïve technique. In practice, even a moderately trained surrogate model yields a tighter reachable set than the naïve approach, which is consistent with our experimental observations and we have provided a justification here for your review.
>
> **Supporting Justification:**
> In the naïve technique, the outputs collected in the training set are approximated using their mean-value, and conformal inference is applied to the prediction errors, resulting in a hyperrectangular reachable set. In contrast, the surrogate-based method fits a small ReLU network to better approximate the outputs. One can, for example, initialize the surrogate model to output a constant equal to the mean of the training data in the beginning of the training process. This initialized model performs exactly as the naive technique and results in the same hyper-rectangular reachset. They also can use **quantile regression**—specifically, the $\delta_1$-quantile of prediction errors—as the loss function. Regardless of how well the surrogate is trained, the quantile of prediction errors will be **strictly lower** than in the initialized model. This ensures that the resulting reachable set is tighter than that produced by the naïve method.
>
> However, due to the high output dimensionality of SSNs, directly training the surrogate without dimensionality reduction was computationally infeasible due to gradient-related challenges. PCA was therefore used to reduce the dimensionality of the output space, enabling us to start training the surrogate model and reduce the quantile of the prediction errors. Thus we achieve a tighter probabilistic reachable set that has the same level of guarantee with the naive technique.
>
> ---
>
> **W (Clarity):** Algorithm 1, in its current form on the submission, is implemented as a for-loop with a fixed number of iterations $N$, and therefore terminates automatically after $N$ steps.
>
> ---
>
> **W (Significance):**
> As suggested by Reviewer 2, we conducted new experiments comparing our method with two recent works that perform probabilistic verification via randomized smoothing. Specifically, we compared our technique with these methods on the verification of an **HRNetV2 model with a W48 backbone** (66M parameters) trained on the **Cityscapes** dataset. Our numerical results show that our approach is approximately **100 times less conservative** than the baselines.
>
> Details of this experiment are provided in our response to Reviewer 2, and we would greatly appreciate it if you could take a look. We have also cited these works in the **Related Work** section of the updated manuscript.
>
> ---
>
> **W (Originality):** As noted, the core techniques used in our framework—conformal inference, surrogate modeling, and PCA—are individually well-established. However, our contribution lies in their novel integration and adaptation to the verification of semantic segmentation networks (SSNs), a high-dimensional and structured output setting where existing methods struggle with scalability or conservatism. This combination results in an efficient and practically useful verification algorithm.
>
> ---
> ---
>
> ***Answer to Questions***
>
> Q1:
>
> **Hyper-parameter sensitivity:**
>
> ---
>
> Q2:
>
> **Dataset coverage:** Our verification algorithm is architecture-agnostic and should work on both the Cityscapes and ADE20K datasets. As previously mentioned, we conducted a new experiment on the Cityscapes dataset, which has now been included in the updated manuscript. The average runtime for this experiment was 210 seconds, and the model we analyzed had 66 million parameters.
>
> ---
> ---

---

> > ### Comment · Reviewer_BgoC · 2025-08-06
> >
> > Thank you for the detailed rebuttal and the additional experiments.
> >
> > I appreciate your explanation of the PCA procedure, including the stopping criteria, and your clarifications on the other points raised.
> >
> > I will maintain my score.

---

> > > ### Author Response · Authors · 2025-08-06
> > >
> > > Thanks a lot for your positive feedback
> > >
> > > We really appreciate it.

---

### Official Review · Reviewer_PV2T · 2025-07-02

**Clarity:** 3
**Significance:** 3
**Originality:** 3
**Rating:** 5
**Confidence:** 3

**Summary:**

The paper presents a probabilistic verification method, based on conformal inference, which is scalable when used in complex semantic segmentation neural networks and lowers conservativism inherent in traditional methods. Experiments are presented with benchmark datasets such as CamVid and Lung segmentation.

**Questions:**

- Please explain and discuss how the values (confidence of guarantee)  in Table 3 relate to the values shown in Figure 1.
- Explain and discuss the differences in average runtimes especially in CamVid and Lung Segmentation shown in Table 3.

**Ethical Concerns:**

["NO or VERY MINOR ethics concerns only"]

**Final Justification:**

Based on my original review comments and my positive evaluation, and the authors' replies both to myself, as well as to the other reviewers' comments, my rating is accept.

**Limitations:**

Yes

**Paper Formatting Concerns:**

No specific formatting issues.

**Quality:**

3

**Strengths And Weaknesses:**

S
- Good theoretical paper for mitigating conservatism in large scale networks such as SSNs.
- Well written, especially section 3, where the reachability technique for robustness verification of SSNs is developed.

W
- It would be good to see more and detailed experimental results (confidence, runtime) in other real life applications, such as the ones represented by CamVid and Lung segmentation.
- It would be good to explore a bit more the way in which the probabilistic framework could be used in safety-critical systems.

---

> ### Author Rebuttal · Authors · 2025-07-30
>
> We appreciate your detailed feedback
>
> ---
> ---
>
> ***Response to Reviewer’s Comments on Weaknesses***
>
> **Extension of Experiments:** We have significantly expanded the experimental section in the revised version. Reviewer 2 also suggested comparisons with two additional methods, which we have now included. The new comparison is described in detail in our response to Reviewer 2, and we would greatly appreciate it if you could take a moment to review that result as well. Below is a summary of the new experimental additions:
>
> 1. In the original submission, parts of the verification process were executed on CPU. We have now migrated all experiments fully on GPU to improve efficiency. For example, our updated experiments on the CamVid dataset now complete in 14 minutes—significantly faster than the 60-minute runtime previously reported.
> 2. We added a new experiment using a different UNet model trained on the OCTA-500 dataset and we analyzed it on a darkening attack like the other experiments in RQ1.
> 3. While the original submission used a fixed perturbation size with varying perturbation dimensions (ranging from $17 \times 3$ to $102 \times 3$) in RQ1, we now include an additional experiment in RQ1 where the perturbation dimension is fixed and the perturbation size is varied from $3/255$ to $150/255$.
> 4. We conducted a conservatism analysis by comparing our reachable set bounds against a Monte Carlo–estimated lower-bound.
> 5. Following the suggestion of reviewer 2, we performed a new comparative study with two other methods on the **Cityscapes** dataset, using an **HRNetV2-W48** backbone with 66 million parameters.
> 6. We extended the experiment in RQ4 (Appendix E, page 14) to cover a broader range of perturbation sizes ($e = 1/255, \ldots, 30/255$) using 200 test images.
> 7. We added several visualizations of verification results across different datasets to improve interpretability.
> 8. We increased the number of test images per experiment to 200 to strengthen statistical confidence in our robustness evaluations.
>
> **Application in Other Domains:** Thank you for this valuable suggestion. We agree that our probabilistic framework has a potential for application in various scientific and engineering domains. In scenarios where deterministic guarantees are difficult or impossible to obtain, our method can be applied and offer  probabilistic but strong guarantees that support reliable decision-making under uncertainty. We believe this is particularly relevant for safety-critical settings and plan to explore such applications further in future work.
>
> ---
> ---
>
> **Answer to Questions**
>
> Q1:
>
> Thank you for the question. Below, we first explain how the Robustness Value (RV) is computed and then clarify how it relates to the Table3 e.g. the hyper-parameters $\ell,m$ and the coverage and confidence $\delta_1, \delta_2$.
>
> Let us begin by describing the process for computing RV:
>
> Assume that $\mathbf{S}$ is the reachable set obtained using our conformal inference method. When we input this reachable set into **Algorithm 2** (Page 8), the output mask is partitioned into three categories: **robust**, **non-robust**, and **unknown** pixels. The **Robustness Value (RV)** is then defined as the fraction of all pixels that are classified as **robust**:
>
> $$
> \text{ RV } = \frac{\text{Number of robust pixels}}{\text{Total number of pixels}}
> $$
>
> Now, let’s relate this to the parameters in Table 3. Suppose we perturb an image from the CamVid dataset over a perturbation set $\mathbf{I}$, we sample a calibration set of size $m = 8000$, and set the rank to be $\ell = 7999$. Using this configuration, we generate a reachable set $\mathbf{S}$ via conformal inference as explained in the paper. We can now define the following logical statement $\mathsf{P}$ based on a coverage level $\delta_1$:
>
> $$
> \mathsf{P} :=  \text{"} \Pr[y \in \mathbf{S}] > \delta_1 \text{"}
> $$
>
> Here, $y$ denotes the vector of logits for all pixels in the SSN output that can be generated from perturbed images in $\mathbf{I}$. The confidence of guarantee, $\delta_2$, then implies:
>
> $$
> \Pr[ \mathsf{P} = \text{True}] > \delta_2
> $$
>
> where since $\mathbf{S}$ is generated via conformal inference using a pre-defined configuration ($ell,m$) we can say,
>
> $$
> \delta_2 = 1 - \mathsf{betacdf}_{\delta_1}(\ell, m + 1 - \ell)
> $$
>
> For example:
> * If we set $\delta_1 = 0.995$, then $\delta_2 = 1 - \mathsf{betacdf}_{0.995}(7999, 2) = 0.99999999999999994$
> * If we set $\delta_1 = 0.999$, then $\delta_2 = 1 - \mathsf{betacdf}_{0.999}(7999, 2) = 0.997$
> * If we set $\delta_1 = 0.9995$, then $\delta_2 = 1 - \mathsf{betacdf}_{0.9995}(7999, 2) = 0.9085$
>
> Using the pair $(\delta_1, \delta_2) = (0.999, 0.997)$, we can formally state that:
>
> $$
> \Pr\big[ \underbrace{\Pr[y \in \mathbf{S}] > 0.999}_{\mathsf{P:=}} \big] > 0.997
> $$
>
> Now suppose we input this reachable set $\mathbf{S}$ into Algorithm 2 and find that 691,100 pixels are classified as **robust**, 100 as **unknown**, and 0 as **non-robust**. The RV is then computed as:
>
> $$
> \text{RV} = \frac{691{,}100}{720 \times 960} \approx 99.98\%
> $$
>
> This means the guarantee
>
> $$
> \Pr\left[ \Pr[y \in \mathbf{S}] > 0.999 \right] > 0.997
> $$
>
> translates directly into the guarantee:
>
> $$
> \\Pr\\left[ \\Pr[\\text{RV} = 99.98\\%] > 0.999 \\right] > 0.997
> $$
>
> In other words, verifying the former implicitly verifies the latter. Therefor, if we generate a reachable  $\mathbf{S}$, with that provable level of guarantee and introduce it to algorithm 2 and find out 691,100 pixels are robust then we have verified the correctness of the following guarantee:
>
> $$
> \\Pr\\left[ \\Pr[\\text{" RV is }  99.98 \\% \\text{" } ] > 0.999 \\right] > 0.997
> $$
>
> Q2:
>
> Thank you for your question. Below, we explain the factors that influence the average runtimes, particularly in the CamVid and Lung Segmentation experiments shown in Table 3.
>
> **Factors Affecting Runtime:**
>
> When using the **naïve technique**, the runtime is primarily influenced by four factors:
>
> 1. The **size of the calibration dataset** $m$,
> 2. The **size of train dataset** $t$,
> 3. The **inference time** of the model, and
> 4. The **memory footprint** on each output vector on the GPU or CPU memory.
>
> The rest of the computation—such as statistical post-processing—has a negligible impact on runtime.  However, inference speed can vary significantly depending on how it is executed. Performing inference on a **GPU with batch processing** is considerably faster than using a **CPU with parallel processing**. Additionally, we collect a training dataset to compute normalization factors used in the definition of the nonconformity score, and this data collection time must also be accounted for.
>
> When using the **surrogate-based technique**, all the factors mentioned above still apply, but there are additional components that impact runtime:
>
> * The time required to compute the **top principal directions** (PCA),
> * The time to **train the surrogate model**, and
> * Although relatively minor, the time required for **deterministic reachability analysis** on the surrogate model (since the surrogate is intentionally kept small and efficient).
>
> **Why CamVid Has Higher Runtime Than Lung Segmentation:**
>
> The runtime for the CamVid dataset is noticeably higher than for Lung Segmentation due to several factors:
>
> * **Model complexity:** CamVid uses **BiSeNet**, which has a more complex architecture and higher inference time compared to **UNet**, used in Lung Segmentation.
> * **Output dimensionality:** The output mask of UNet in the Lung Segmentation task is significantly smaller than that of BiSeNet in CamVid. Smaller outputs reduce memory requirements for each vector and enable more efficient batch processing during inference. As a result, we can process larger batches in the Lung Segmentation setting, which leads to a substantial reduction in overall verification time.
>
> **Additional Notes:**
>
> * The **verification runtime is independent of perturbation size**. That is, verifying for $e = 3/255$ takes approximately the same amount of time as verifying for $e = 155/255$.
> * In the updated version of the paper, we include a chart in the appendix visualizing the runtime across all experiments introduced in RQ1.
> * Although increasing the perturbation size $e$ does not increase runtime, it does increase the conservatism of the reachable set if the size of training dataset is not also increased. While the probabilistic guarantee still holds in such cases, the conservatism may become more pronounced. Thus we suggest increasing the size of training dataset if the perturbation size is larger.
>
> ---
> ---

---

> ### Comment · Reviewer_PV2T · 2025-08-06
>
> Dear authors: Thank you for the rebuttal and the detailed responses provided to the mentioned weaknesses and questions. I will retain my score.

---

> > ### Author Response · Authors · 2025-08-06
> >
> > Dear Reviewer
> >
> > Thanks a lot for your positive feedback
> >
> > We really appreciate it.

---

### Official Review · Reviewer_uuuW · 2025-07-02

**Clarity:** 3
**Significance:** 3
**Originality:** 3
**Rating:** 5
**Confidence:** 4

**Summary:**

The paper proposes a new certification method to prove the robustness of semantic segmentation models against adversarial perturbations. In contrast to precise methods based on reachability analysis, the authors propose to instead compute probabilistic guarantees using conformal inference. This is inspired by the high output dimensionality of segmentation models (one classification per pixel), which makes them more expensive to certify. The approach is validated on segmentation models up to ResNet18-size for CamVid, Lung_Segmentation, and M2NIST datasets.

**Questions:**

1. Is there any way to get results on Cityscapes or ACDC? Since you mention self-driving cars as motivation, this would complete the paper, make it relevant to practical datasets, and comparable to prior work

2. Can you please compare to the related work listed above [a, b, c, d] - both conceptually and empirically?

3. L53: what do you mean by "low-conservatism"?

4. L292: In 2025, a 64x64 pixel dataset cannot really be described as "high-dimensional". I understand that the method has limited scalabliliy in terms of image resolution, but please be upfront about it.

5. L294: Same for "large and complex pre-trained models" - the largest mode lis ResNet18, which in 2025 cannot be considered large or complex.

6. L297: What is the reason for this particular threat model? Why is the method especially applicable in this setting?

**Ethical Concerns:**

["NO or VERY MINOR ethics concerns only"]

**Final Justification:**

The authors addressed all of my major concerns during the rebuttal. In particular, they clarified many details and provided additional results and comparisons to prior work. I therefore recommend acceptance with the following changes, to which the authors agreed:

- The additional results and comparisons to related works (including the extended [b] and [d] table and the Cityscapes/HRNetV2 experiments) are included in the paper.
- The clarifications and additional details from the author responses are integrated, including the distinction between the naive and surrogate methods, and the justification for the chosen threat models.
- The major limitations are explicitly discussed in a limitations section, including (1) the distributional assumption, which may not hold in an adversarial setting, (2) the requirement of recomputing the calibration set for each test point, and (3) the scalability limitations wrt the input dimension of the surrogate approach.

**Limitations:**

The paper does not address limitations, even though there are many. For example, guarantees are only probabilistic and could fail; it is limited to very small models and datasets; Evaluation does not compare to any baselines; the method is not applicable to the real-time requirements of some of the motivating examples (e.g., self-driving cars); etc.

**Paper Formatting Concerns:**

Some tables and figures are tiny (e.g. Tables 2, Figure 1) which make them impossible to read without zooming (e.g. on print).

**Quality:**

2

**Strengths And Weaknesses:**

**Strengths**

1. It has been well established that neural networks - and semantic segmentation models in particular - are vulnerable to small perturbations. Provable worst-case guarantees are therefore highly desired, which makes this work a very valuable contribution.

2. To the best of my knowledge, the approach is novel in the context of semantic segmentation. While scalability and practical applicability currently seem limited (see below), I believe that the space of certified robustness methods still requires significant exploration. I therefore appreciate the novel perspective on certified robustness guarantees.

3. The mathematical guarantees appear sound to me, although I am not an expert on conformal predictions and statistics, and therefore may have missed subtle problems.

**Weaknesses**

1. The paper lacks comparisons to prior work and baselines, both in related works and experiments. There are at least two lines of work that can certify the robustness of segmentation models: (i) based on sound reachability analysis [a], and (ii) based on randomized smoothing, which also computes probabilistic certificates [b, c, d]. Even if the proposed approach does not outperform SOTA methods, it would be important to understand the differences in the threat model, precisely what is guaranteed, and the performance in terms of certified accuracy, robust radius, and model and dataset sizes.

2. The evaluation is limited in several aspects:
 - The evaluation is limited to 10/20 test data points. This is insufficient to reach statistically relevant results. Based on the runtimes in Table 3, it should be feasible to run the experiments on at least ~500 - 1000 datapoints in a few days. (2)
 - The threat model considered is non-standard for the adversarial robustness literature. Typically, prior work assumes either $\ell_p$-norm perturbations or patch attacks. Why was this particular threat model chosen? If sparsity is a requirement, $\ell_0$ perturbations would be the standard setting that makes the work comparable to prior work. Otherwise, there should be a discussion and justification of the particular threat model.

3. The authors do not provide code for evaluation and extension of the proposed method.


**Conclusion**

In its current form, the paper's weak evaluation and lack of comparison to prior work outweigh its significant contribution and interesting approach. If these two points are addressed, I would be willing to change my rating to acceptance, even if the results fall behind SOTA.


[a] Tran, HD. et al. (2021). Robustness Verification of Semantic Segmentation Neural Networks Using Relaxed Reachability. In: Silva, A., Leino, K.R.M. (eds) Computer Aided Verification. CAV 2021
[b] Scalable Certified Segmentation via Randomized Smoothing. Marc Fischer, Maximilian Baader, Martin Vechev Proceedings of the 38th International Conference on Machine Learning, 2021
[c] GSmooth: Certified Robustness against Semantic Transformations via Generalized Randomized Smoothing
Zhongkai Hao, Chengyang Ying, Yinpeng Dong, Hang Su, Jian Song, Jun Zhu Proceedings of the 39th International Conference on Machine Learning, 2022
[d] Adaptive Hierarchical Certification for Segmentation using Randomized Smoothing. Alaa Anani, Tobias Lorenz, Bernt Schiele, Mario Fritz Proceedings of the 41st International Conference on Machine Learning, 2024

---

> ### Author Rebuttal · Authors · 2025-07-30
>
> We appreciate your detailed feedback.
>
> ---
> ---
>
> ***Response to Reviewer’s Comments on Weaknesses***
>
> Comparison with SOTA:
>
> We were aware of paper [a]. In RQ2 (Appendix E, page 13), we showed NNV encountered out-of-memory issues in all experiments presented in RQ1. However, we were not previously aware of the other three papers [b, c, d], and we have now added them to related works and also have a comparison with them in the updated paper. The guarantee formulations are slightly different from ours. However, we found a meaningful way to compare, and our numerical results show our technique is 100 times less conservative. We also provide the details of this comparison below for your review.
>
> **Overview:**  Papers [b, c, d] are based on randomized smoothing. Paper [b] introduces a method called $\mathsf{SEGCERTIFY}$.  Lets assume  $\overline{\mathsf{SSN}}(x)(i,j)$ is the smooth version of $\mathsf{SSN}(x)(i,j)$ on pixel $(i,j)$, generated by a zero mean gaussian random noise:
> $$
> \overline{\mathsf{SSN}}(x)(i,j)=c_A(i,j)=\underset{c\in\mathbf{L}}{\arg\max}\Pr_{\nu\sim\mathcal{N}(0,\sigma^2)}\left[\mathsf{SSN}(x+\nu)(i,j)=c\right]
> $$
> where $\mathbf{L}$ is the set of classes. Assuming a coverage level $\tau\in[0.5,1]$, they redefine the top class $c_A(i,j)$ in a more conservative yet practically more convenient manner, and specify it as the class that satisfies:
> $
> \Pr_{\nu\sim\mathcal{N}(0,\sigma^2)}\left[\mathsf{SSN}(x+\nu)(i,j)=c_A(i,j)\right]\ge\tau
> $
> , and then given a confidence guarantee $\delta_2$, they then propose the following guarantee for the smooth approximation of the model:
> $$
> \forall x'\in\mathbf{B}_r(x):\Pr\left[\overline{\mathsf{SSN}}(x')(i,j)=c_A(i,j)\right]\ge\delta_2
> $$
> where $\mathbf{B}_r(x)$ is a ball with radius $r=\sigma\Phi^{-1}(\tau)$ and center $x$. They also argue that by allowing a portion of pixels to be abstained from (i.e., remain uncertifiable), they  define:
> $$
> \mathbf{CERT}=\\{(i,j)|(i,j)\mathsf{\ is\ certifiable}\\}
> $$
> and extend this guarantee over output space of $\mathsf{SSN}$ as:
> $$
> \\forall x'\\in\\mathbf{B}\_r(x):\\Pr\\big[\\bigwedge\_{(i,j)\\in\\mathbf{CERT}}\\overline{\\mathsf{SSN}}(x')(i,j)=c_A(i,j)\\big]\\geq\\delta_2
> $$
> which can be rephrased for the base model as:
>
> $$
> \\forall x'\\in\\mathbf{B}\_r(x):\\Pr\\big[\\bigwedge\_{(i,j)\\in\\mathbf{CERT}}\\Pr_{\\nu\\sim\\mathcal{N}(0,\\sigma^2)}\\left[\\mathsf{SSN}(x'+\\nu)(i,j)=c_A(i,j)\\right]\\ge\\tau\\big]\\ge\\delta_2
> $$
> On the other hand, the authors in [d] enhance this work using adaptive techniques to reduce the number of abstained (uncertifiable) pixels, and call it $\mathsf{ADAPTIVECERTIFY}$.
>
> **Comparison:**  We adopt the perturbation set $\mathbf{B}_r(x)$ identical to what considered above and present our guarantee for certifying the robustness of the output pixels. Due to the conservative nature of our reachability analysis, some bounds for certain mask pixels may span multiple classes, which we label as unknown. These unknown pixels are in fact uncertifiable, and thus we define the following set accordingly:
> $$
> \mathbf{Conf\\_CERT}=\\{(i,j)|(i,j)\mathsf{\ is\ certifiable}\\}
> $$
> and given a coverage level $\delta_1 $ and confidence $\delta_2 $ and a sampling distribution, $x'\stackrel{\mathcal{W}}{\sim}\mathbf{B}_r(x)$, we generate the following guarantee:
>
> $$
> \\mathbf{If}\\ x'\\stackrel{\\mathcal{W}}{\\sim}\\mathbf{B}\_r(x):\\ \\  \\Pr\\big[\\Pr\\big[\\bigwedge\_{(i,j)\\in\\mathbf{Conf\\_CERT}}\\mathsf{SSN}(x')(i,j)=\\mathsf{SSN}(x)(i,j)\\big]\\ge\\delta_1\\big]\\ge\\delta_2
> $$
> Therefore, given an identical perturbation set $\mathbf{B}_r(x)$ the comparison aims to determine which method produces fewer uncertifiable pixels under its proposed guarantee.  We use Table 1 from [d] as a benchmark, which sets $\tau=0.75,0.95$, $\sigma= 0.25,0.33,0.5$, and $\delta_2=0.999$ (6 experiments totally, 200 images each). Their experiments are conducted using this configuration on an HRNetV2 model with a W48 backbone, trained on the **Cityscapes** dataset, and they compare the number of uncertifiable pixels achieved against [b]. Accordingly, we use our naive technique and extend this table by reporting the number of uncertifiable pixels we obtained under the setting $\delta_1=0.99,\delta_2=0.999$ and $\mathcal{W}$ to be uniform for all of those 6 experiments. The code is available online anonymously and can be shared upon your request during the author-reviewer discussion period.
>
> **Results**:  The number of uncertifiable pixels produced by our technique was 100 times smaller than the values reported in that table, **clearly demonstrating the effectiveness of incorporating conformal inference into the verification of SSNs**.  Our average runtime was 210 seconds, with a calibration dataset size of $m=920$ for all experiments. However, Table 1 in [d] does not report any runtime information.
>
> ---
>
> Limitation in Evaluation:
>
> In response to the reviewer's concerns, we have significantly expanded and improved our evaluation. Specifically, inspired by prior works \[b, d], we increased the number of test images to 200 across all experiments and added several new evaluations, as detailed below:
>
> 1. In the original submission, some parts of verification were conducted on CPU; all experiments are now executed solely on GPU for improved efficiency.
> 2. We added a new experiment on another UNet trained on the OCTA-500 dataset.
> 3. While the original submission focused on a fixed perturbation size with varying perturbation dimensions (ranging from $17\times 3$ to $102\times 3$), the revised version includes an additional experiment where we fix the perturbation dimension and vary the perturbation size from $3/255$ up to $150/255$.
> 4. We included analyzing the conservatism of our method by comparing our reachable set bounds to a lower-bound estimated via Monte Carlo simulation.
> 5. Per your suggestion, we performed a new comparative study with \[b, d] on the **Cityscapes** dataset, targeting an **HRNetV2-W48** backbone with 66M parameters.
> 6. We extended the experiment in RQ4 (Appendix E, page 14) to a broader range of perturbation sizes ($e = 1/255, ..., 30/255$), using 200 test images.
> 7. We included several visualizations of our verification results across different datasets to enhance interpretability.
>
> Regarding the threat model, we have incorporated both:
>
> * $\ell_\infty$ perturbations were used in the original RQ3 and RQ4 experiments on the CIFAR-100 dataset (Appendix E).
> * A new experiment on the **Cityscapes** dataset, in comparison with \[b, d], uses $\ell_2$ perturbations.
>
> Our primary motivation for using the darkening attack was also as follows:
>
> 1. It was the primary threat model considered in \[a], and widely used in several other papers on adversarial training.
> 2. It introduces sparsity in the input space.
> 3. It allows easy control over both perturbation size and dimension, enabling us to thoroughly test the scalability of our technique.
>
> Finally, the code and updated manuscript are available online via an anonymous link. We are happy to share access upon your request during the author-reviewer discussion period.
>
> ---
> ---
>
> ***Answer to the Questions***
>
> Q1:
>
> Following your suggestion, we conducted experiments on the Cityscapes dataset to compare our results with methods [b] and [d].
>
> Q2:
>
> We previously compared our method with [a] in RQ2 (Appendix E). Method [d], as an extension of [b], offered a coherent narrative for comparison, and we included both conceptual and empirical comparisons against [b,d]. In contrast, [c] focuses on transformations with a focus on classification tasks, and we only cited that in Related works.
>
> Q3:
>
> We revised this sentence as follows:
> " In this work, we propose a scalable, architecture-agnostic,  probabilistic verification algorithm tailored for semantic segmentation neural networks and we demonstrate through numerical experiments that it is less conservative than existing statistical methods in the literature.".
>
> Q4:
>
> We have addressed this by removing the mention of "M2NIST" from the sentence. One comment here is:
>
>  * Our naïve approach remains scalable to high-resolution input images for SSN, as it does not require training a surrogate model. For example, in our comparison with [b, d], the naïve technique completes verification in 210 seconds for an image with a resolution of 1024×2048.
> * As noted in Lines 269–274, when the SSN input is high-dimensional, our surrogate-based method remains efficient under the assumption of sparsity in the input perturbation.
>
> Therefore, in cases where the perturbation dimension is very large, we fall back to the naïve technique for scalability.
>
> Q5:
>
> We agree with your observation. Like methods [b, c, d], our approach is architecture-agnostic and should generalize to larger models. While the models used in RQ1 were not representative of large-scale architectures, as you suggested, we have now added an experiment using the HRNetV2-W48 backbone—which has 66 million parameters—to compare our technique with methods [b,d].
>
> Q6:
>
> Our goal in RQ1 was mainly to evaluate the surrogate-based technique. While PCA enables the surrogate method to scale to SSNs with high-dimensional outputs, it does not address scalability when the input dimensionality is high. As noted in the manuscript, the surrogate approach requires sparsity in the perturbation set when dealing with high-dimensional inputs. This motivated our choice of the darkening attack, which introduces structured, sparse perturbations.
>
> ---
> ---
>
> ***Response to Limitation Comments***
>
> * We clearly stated in the conclusion section that our method provides probabilistic guarantees, and we also emphasized on this in the title of the paper.
> * Regarding model size, our approach is not limited to small models—we successfully performed verification on HRNetV2 with 66 million parameters in just 210 seconds, achieving significantly lower conservatism compared to methods [b] and [d] on the Cityscapes dataset.
>
> ---
> ---

---

> > ### Author Response · Authors · 2025-08-03
> >
> > Dear Reviewer uuuW,
> >
> > Thank you again for your detailed and constructive feedback. Based on your suggestion, we have conducted the comparison with the relevant prior work as you recommended. The comparison results have been added in the discussion above.
> >
> > We would greatly appreciate it if you could take a moment to review the new comparison and share your thoughts. Your feedback would be very valuable to us, especially given your earlier note that addressing this point could potentially change your overall recommendation.
> >
> > Thank you again for your time and consideration.
> >
> > Best regards,

---

> > > ### Author Response · Authors · 2025-08-06
> > > **Request for your feedback**
> > >
> > > Dear Reviewer uuuW,
> > >
> > > Thank you again for your detailed and constructive feedback. Based on your suggestion, we have conducted the comparison with the relevant prior work as you recommended. The comparison results have been added in the discussion above.
> > >
> > > We would greatly appreciate it if you could take a moment to review the new comparison and share your thoughts. Your feedback would be very valuable to us, especially given your earlier note that addressing this point could potentially change your overall recommendation.
> > >
> > > Thank you again for your time and consideration.
> > >
> > > Best regards,

---

> ### Comment · Reviewer_uuuW · 2025-08-06
>
> Dear Authors,
>
> Thank you for the detailed response and for the new experiments on Cityscapes and HRNetV2. I appreciate the significant improvements in evaluation and the more thorough discussion of comparisons.
>
> I have a few clarifications and follow-up questions:
>
> **1. Quantitative Comparison**
>
> You mention that your method produces “100× fewer uncertifiable pixels” than [d]. According to Table 1 in [d], [b] has between 5% and 28% abstained pixels. Does this imply your method abstains from only 0.05% to 0.28% of pixels for the same perturbation settings?
>
> Could you please provide the extended table you mentioned that directly compares the methods?
>
> More generally, randomized smoothing provides stronger guarantees—i.e., worst-case robustness over a norm ball—without relying on assumptions about the distribution of perturbations. Would it be fair to say that the lower abstention rates in your method come at the cost of weaker guarantees, making this more of a trade-off than a superiority?
>
> **2. Perturbation Sparsity**
>
> You mention that perturbation sparsity is essential for your surrogate-based method to scale to high-dimensional inputs. Yet the comparison on Cityscapes uses $\ell_2$ perturbations, which are dense by nature. Could you clarify whether the surrogate method was used in this case, and if so, how it could scale to dense perturbations on high-dimensional inputs?
>
> **3. Calibration Set**
>
> Do you reuse the same calibration set across all test examples (e.g., the 200 Cityscapes images), and if so, how do you account for potential overfitting or leakage when estimating coverage guarantees?

---

> > ### Author Response · Authors · 2025-08-06
> > **Thanks for your feedback**
> >
> > **Dear Reviewer,**
> >
> > We appreciate your detailed feedback and thank you again for introducing those papers to us.
> >
> > Below, we provide detailed responses to your questions.
> >
> > ---
> >
> > **Q1:**
> >
> > Yes, our technique resulted in uncertifiable pixels within a lower range (less than 0.05%–0.28%), as reported for the exact same perturbation set $\mathbf{B}_r(x)$. We have provided the extended table here for your review.
> >
> > 1- $\sigma = 0.25, \tau = 0.75$: ($ r = 0.1686$), the percentage of uncertifiable pixels from [b],[d] and our naive approach are, 7%, 5%, 0.0642% repectively.
> >
> > 2- $\sigma = 0.33, \tau = 0.75$: ($ r = 0.2226$), the percentage of uncertifiable pixels from [b],[d] and our naive approach are, 14%, 10%, 0.0676% repectively.
> >
> > 3- $\sigma = 0.50, \tau = 0.75$: ($ r = 0.3372$),  the percentage of uncertifiable pixels from [b],[d] and our naive approach are, 26%, 15%, 0.0705% repectively.
> >
> > 4- $\sigma = 0.25, \tau = 0.95$: ($ r = 0.4112$),  the percentage of uncertifiable pixels from [b],[d] and our naive approach are, 12%, 9%,  0.0727% repectively.
> >
> > 5- $\sigma = 0.33, \tau = 0.95$: ($ r = 0.5428$),  the percentage of uncertifiable pixels from [b],[d] and our naive approach are, 22%, 18%, 0.0732% repectively.
> >
> > 6- $\sigma = 0.50, \tau = 0.95$: ($ r = 0.8224$), the percentage of uncertifiable pixels from [b],[d] and our naive approach are, 39%, 28%, 0.0758%  repectively.
> >
> > We agree that randomized smoothing considers a worst-case distribution, whereas our technique provides a guarantee under a prior distribution. However, randomized smoothing offers guarantees only for the smoothed model, not the base model, whereas our technique directly targets the base model.
> >
> > If we attempt to rephrase the guarantee on the smoothed model,
> >
> > $$
> > \\forall x'\\in\\mathbf{B}\_r(x):\\Pr\\big[\\bigwedge\_{(i,j)\\in\\mathbf{CERT}}\\overline{\\mathsf{SSN}}(x')(i,j)=c_A(i,j)\\big]\\geq\\delta_2
> > $$
> >
> > to apply it to the base model, the guarantee takes the following form:
> >
> > $$
> > \\forall x'\\in\\mathbf{B}\_r(x):\\Pr\\big[\\bigwedge\_{(i,j)\\in\\mathbf{CERT}}\\Pr_{\\nu\\sim\\mathcal{N}(0,\\sigma^2)}\\left[\\mathsf{SSN}(x'+\\nu)(i,j)=c_A(i,j)\\right]\\ge\\tau\\big]\\ge\\delta_2
> > $$
> >
> > This implies that a prior Gaussian distribution is also involved in their guarantee. If we attempt to remove this noise (i.e., consider \$\nu = 0\$ to analyze the base model directly), then \$\sigma = 0\$, which implies \$r = \sigma \Phi^{-1}(\tau) = 0\$, making it impossible to provide a meaningful guarantee.
> >
> > In contrast, our technique directly considers the base model without any added noise \$\nu\$ to \$x'\$:
> >
> > $$
> > \\mathbf{If}\\ x'\\stackrel{\\mathcal{W}}{\\sim}\\mathbf{B}\_r(x):\\ \\  \\Pr\\big[\\Pr\\big[\\bigwedge\_{(i,j)\\in\\mathbf{Conf\\_CERT}}\\mathsf{SSN}(x')(i,j)=\\mathsf{SSN}(x)(i,j)\\big]\\ge\\delta_1\\big]\\ge\\delta_2
> > $$
> >
> > Again, we acknowledge your point: while randomized smoothing assumes a worst-case distribution, our method considers a prior distribution \$\mathcal{W}\$. In future work, we plan to refine the choice of \$\mathcal{W}\$, possibly incorporating adversarial directions such as PGD and FGSM in the sampling process.
> >
> > In conclusion, we fully agree with your observation that the much lower abstention rate reported for our method stems from considering a prior distribution. **We do not view this as an advantage, but rather as part of a trade-off between the two approaches, not a sign of superiority**.
> >
> >
> > **Q2:**
> >
> > Due to the high dimensionality and nature of the perturbation set, we utilized our naive technique for this comparison. The surrogate method did not scale to this experiment for two key reasons:
> >
> > 1. **High Input Dimensionality:** The input’s high dimensionality made the training process for the surrogate model difficult. As noted in the paper, our dimensionality reduction techniques (e.g., learning-based PCA) only address output space dimensionality, not input space. Our surrogate method requires sparsity in the perturbation, which is not a requirement for our naive method.
> >
> > 2. **Perturbation Geometry:** The perturbation is $\ell\_2$, corresponding to a sphere in $\mathbb{R}^n$. The surrogate method assumes a star set for deterministic reachability, which requires bounding this sphere with a hypercube. However, the hypercube would be $\sqrt{n}$ times larger than the sphere, making the surrogate approach infeasible  for large $n$ due to the large overapproximation.
> >
> > Given these limitations, we accepted a higher level of conservatism and opted to use our naive technique for comparisons with [b, d].
> >
> > **Q3:**
> >
> > The calibration set is collected by sampling within the perturbation bounds for each image. This means that for every new image used in computing the SSN’s robustness, a new calibration dataset must be collected.
> >
> > The runtime of 210 seconds we reported includes also the time for collecting the calibration dataset. We have provided a more detailed explanation of the factors affecting verification runtime in our response to Reviewer 3.

---

> > > ### Author Response · Authors · 2025-08-07
> > > **Deadline is Extended**
> > >
> > > Dear Reviewer,
> > >
> > > We wanted to inform you that the deadline for the author–reviewer discussion period has been extended to August 8th.
> > >
> > > Thank you for your time and engagement.
> > >
> > > Best regards,

---

> > > ### Comment · Reviewer_uuuW · 2025-08-08
> > >
> > > Dear authors,
> > >
> > > Thank you for the replies and additional clarifications. I would be in favor of accepting the paper, provided:
> > >
> > > - The additional results and comparisons to related works (including the extended [b] and [d] table and the Cityscapes/HRNetV2 experiments) are included in the paper.
> > > - The clarifications and additional details from the author responses are integrated, including the distinction between the naive and surrogate methods, and the justification for the chosen threat models.
> > > - The major limitations are explicitly discussed in a limitations section, including (1) the distributional assumption, which may not hold in an adversarial setting, (2) the requirement of recomputing the calibration set for each test point, and (3) the scalability limitations wrt the input dimension of the surrogate approach.
> > >
> > > I will adjust my rating accordingly and trust that the authors will take the opportunity to refine the paper.
> > >
> > > PS: I am aware of the reviewing and discussion timeline and would appreciate it if we could keep the discussion within its intended scope, without repeated reminders.

---

> > > > ### Author Response · Authors · 2025-08-08
> > > > **Thanks for the acceptance**
> > > >
> > > > Dear Reviewer,
> > > >
> > > > Thank you very much for your constructive feedback, for outlining the specific points for improvement, and for your support toward acceptance.
> > > >
> > > > We will ensure that the final version of the paper fully incorporates all three points you mentioned exactly.
> > > >
> > > > Thank you again for your time and valuable input.

---

### Official Review · Reviewer_Xbuv · 2025-07-03

**Clarity:** 4
**Significance:** 2
**Originality:** 2
**Rating:** 3
**Confidence:** 2

**Summary:**

This paper addresses the challenge of verifying robustness in Semantic Segmentation Neural Networks (SSNs), particularly under input perturbations and adversarial attacks, where formal verification approaches are often computationally intractable due to the high-dimensional output space of SSNs. The authors propose a scalable, architecture-agnostic probabilistic verification framework that leverages conformal inference (CI) for data-driven reachability analysis with formal probabilistic guarantees.

A naive baseline is introduced that applies CI to SSNs by constructing hyper-rectangular reachsets, which are provably correct but overly conservative. The authors then improve this baseline by: 1. Incorporating surrogate ReLU networks to approximate complex SSNs, 2. Using zonotopic (star-set) representations instead of axis-aligned boxes, 3. Applying a scalable Principal Component Analysis (PCA) technique (via a deflation algorithm) to handle high-dimensional output.

Experiments on benchmark datasets (M2NIST, CamVid, Lung Segmentation) and models (U-Net, BiSeNet, ResNet) demonstrate the practicality of the proposed method, showing its capability to provide provable guarantees with lower conservatism and scalability to large-scale networks.

**Questions:**

1. You assume that a small ReLU network can accurately approximate the SSN under the input distribution. What happens when this approximation fails? Could you provide an error bound or confidence estimate for the surrogate's fidelity?

2. The paper emphasizes that the proposed method is less conservative than the naive baseline, but does not quantify this. Can you provide a metric (e.g., volume ratio, false positive rate) to quantify the reduction in conservatism introduced by surrogate-based starsets over hyper-rectangles?

3. You use learning-based PCA for output compression. Does this technique scale to more complex SSNs with class imbalance or highly correlated outputs (e.g., medical segmentation)? Could more sophisticated dimensionality reduction (e.g., autoencoders) offer advantages?

4. These tasks also produce structured, high-dimensional outputs. What changes would be needed for your approach to work on non-semantic segmentation tasks?

5. The framework relies on users tuning calibration size and miscoverage thresholds. Is there a principled way to choose these values to balance runtime and conservativeness, or are they empirically tuned?

**Ethical Concerns:**

["NO or VERY MINOR ethics concerns only"]

**Limitations:**

Yes (but partially). The paper discusses the limitations of conformal inference in high-dimensional settings and the lack of deterministic guarantees. However, some critical points are underexplored, including: 1. How approximation errors in the surrogate model affect guarantees; 2. Potential failure of the method when calibration distributions are not representative; 3. No discussion of societal impact, e.g., false security in safety-critical applications like medical imaging.

**Paper Formatting Concerns:**

1. Mathematical notation is mostly clear but can occasionally be dense. Some equations would benefit from explanatory text or illustrative examples.
2. Figures (e.g., RV vs. # perturbed pixels) are informative, but axes and legends could be clearer.
3. Tables are helpful but could include more runtime and memory comparison vs. baseline methods.
4. Minor typos: e.g., “deflative PCA” → “deflation-based PCA”; “conservatism” used as a noun could be rephrased.

**Quality:**

3

**Strengths And Weaknesses:**

Strengths

The framework is designed to scale to large segmentation models and high-dimensional output spaces, a major bottleneck in prior work on neural network verification.

The method builds on conformal inference, with sound statistical foundations. The paper offers a formal double-layer probabilistic guarantee, balancing confidence and miscoverage.

The authors start from a transparent, naive CI-based technique and systematically address its limitations via star-set reachability, surrogate modeling, and dimensionality reduction.

The use of learning-based PCA (via deflation) to address high output dimensionality in robustness analysis is innovative and effective.

The experiments target relevant questions (e.g., scalability, conservatism, runtime), are run across diverse datasets and models, and demonstrate superior scalability compared to deterministic methods like αβ-CROWN and NNV.

The formalization of the probabilistic reachsets via ⟨ϵ, ℓ, m⟩ tuples is precise and instructive, making it easier to interpret coverage and confidence.

Weaknesses

The key technical building blocks (conformal inference, surrogate modeling, PCA) are individually well-established. The contribution lies in their combination and adaptation to SSNs rather than a fundamentally new technique.

While the authors emphasize scalability, there’s no formal analysis of conservativeness versus exactness. The surrogate model approximation is only loosely justified, and the inflation method may introduce nontrivial looseness.

Training a ReLU surrogate model assumes a sufficiently accurate approximation of the original SSN. But the conditions under which the surrogate is guaranteed to succeed are not discussed, especially for very deep or complex architectures.

The method is evaluated under a synthetic "darkening" attack and UBAA noise models. More standard threat models (e.g., FGSM, PGD, spatial transformations) are not used, which limits generalizability.

While pixel-level robustness is central, it's not always clear how the user should interpret “unknown” regions in practice—especially when aggregated over classes or images.

---

> ### Author Rebuttal · Authors · 2025-07-29
>
> We appreciate your detailed feedback.
>
> ---
>
> ***Response to Comments on Weaknesses***
>
> **On the Use of Established Techniques:**  As noted, the core techniques used in our framework—conformal inference, surrogate modeling, and PCA—are individually well-established. However, our contribution lies in their novel integration and adaptation to the verification of semantic segmentation networks (SSNs), a high-dimensional and structured output setting where existing methods struggle with scalability or conservatism. This combination results in an efficient and practically useful verification algorithm. As requested by Reviewer 2, we have included additional experiments comparing our method with several baselines from the literature. These results, discussed in our responses to Reviewer 2, demonstrate a significantly reduced level of conservatism in our method relative to existing approaches.
>
> **On Conservatism vs. Exactness:** Appendix E (RQ4) already provides a numerical comparison showing that the ReLU-based surrogate leads to less conservative estimates than the naive baseline. However, you are correct that we did not previously include a numerical evaluation of conservatism relative to exact reach sets. To address this, we have added a new experiment in the revised version of the paper. Specifically, we perform 10^6 random forward passes through the model to empirically lower-bound the true reachable set. We then compare these empirical bounds with the reach set computed by our method. The results show that, on average, the bounds produced by our method are only 10% larger than the empirical lower bound in the M2NIST experiment, and approximately 20% larger in the CamVid experiment. This demonstrates that while our method is conservative, the gap from exactness remains moderate and practical.
>
> **On Assumptions about Surrogate Model Accuracy:** We do not assume an accurate approximation of the original SSN. The main motivation for introducing the surrogate technique is to improve upon the naive method. The more accurate the ReLU surrogate, the lower the conservatism compared to the naive technique. We discuss this in more detail in our responses to your questions.
>
>
> **On the Use of Attack Models:** We acknowledge that FGSM and PGD are widely used in adversarial robustness literature. However,PGD and FGSM attacks are essentially one-dimensional, as they identify a single critical direction to which the model is most vulnerable. Verifying robustness against such attacks does not require reachability techniques. Since pixel values are discrete (e.g., $1/255, 2/255, \ldots, 1$), verification along a one-dimensional perturbation can be efficiently performed with a finite number of forward passes through the model. In contrast, consider a 102-dimensional perturbation with a magnitude of $3/255$. Verifying robustness in this setting would require $3^{102}$ model evaluations, which is computationally infeasible. This complexity motivates the need for reachability-based analysis. For this reason, we focused on darkening attacks in our experiments.
>
> **On the Interpretation of Unknown Pixels:** We agree that pixels labeled as “unknown” (i.e., not certifiably robust) can pose a challenge in interpretation. The unknown pixels are not certifiable, meaning we do not provide guarantees for them due to the conservatism inherent in our reachability analysis. Addressing uncertifiable pixels is a common and active area of research in the verification community. However, as requested by Reviewer 2, we described a recent experiment demonstrating that the number of uncertifiable pixels produced by our method is at least 100 times lower than those reported by other probabilistic techniques in the literature.
>
> ---
>
> ***Answer to the Questions:***
>
> Q1:
>
> We do not assume that a small ReLU network accurately approximates the SSN across the entire input distribution. Instead, we use the network to enhance the naive technique in two ways:
>
> **Improved Output Estimation:** In the naive method, we estimate the outputs using a fixed value (typically the mean), which can result in high approximation error. By contrast, we use a small ReLU model to learn a more expressive approximation of the outputs, which reduces this error. The better the surrogate model, the more accurate the output estimates, resulting in a tighter reachable set.
>
> **Better Reachable Set Geometry:** We apply ApproxStar reachability analysis to the ReLU model, which yields a convex hull over the output space. This provides a more informative and compact approximation of the cloud of outputs than the hyper-rectangular (box-shaped) reachable sets used in the naive approach. To ensure formal coverage guarantees, we then use conformal inference to appropriately inflate this convex hull, ensuring the true outputs lie within it with provable probabilistic guarantee.
>
> In summary, even if the small ReLU model is not a perfect surrogate for the SSN, it still leads to tighter and more meaningful reachable sets than the naive mean-based method.
>
> **Proof:** One can initialize the surrogate model such that it outputs a constant value equal to the mean of the training dataset and define the loss function as the $\delta_1$-quantile of the prediction errors (i.e.,**Quantile Regression**). Regardless of how accurately the surrogate is trained, the resulting error quantile will be lower than that of the initialized model. This is sufficient to claim that the new reachable set has been tightened.
>
> However, due to the high dimensionality of SSN, we were unable to start training it, primarily because of gradient-related computational issues. To address this, we applied PCA to reduce dimensionality, allowing the training process to get started.
>
> Q2:
>
> Yes, we addressed this through a numerical experiment presented in RQ4 of Appendix E (Page 14), where we evaluated the conservatism of our method. The experiment compares three approaches: (1) the naive technique, (2) reachability analysis with a linear surrogate model, and (3) reachability analysis with a ReLU-based surrogate model.
>
> In the original experiment, we varied the perturbation bound $e$ from $1/255$ to $10/255$. The ReLU-based method verified robustness of the classification up to $e = 9/255$, while the linear surrogate method did so up to $6/255$, and the naive method only up to $3/255$. This clearly demonstrates the improved performance of the ReLU surrogate, which we attribute to its ability to produce less conservative reachable sets. We recently extended this analysis to $200$ images and expanded the range of $e$ values from $1/255$ to $30/255$.
>
> Q3:
>
> Before addressing it directly, we would like to highlight that, due to the accuracy and data efficiency of CI, our baseline (naive) technique performs better compared to SOTA. This is supported by new experimental results requested by Reviewer 2, which show it significantly outperforms existing probabilistic methods. However, our overarching goal is to reduce conservatism of naive technique, which is why we incorporate ReLU-based surrogate models in reachability.
>
> We terminate our SGD-based PCA algorithm once a specific amount of principal directions are extracted. When outputs are highly correlated, that amount may be insufficient. This can degrade the accuracy of the ReLU surrogate model. Nonetheless, even in these cases, the ReLU-based reachability analysis still outperforms the naive approach, as learning from the data distribution using a ReLU model is more effective than relying on approximation with a mean-value (naive technique).
>
> We appreciate the suggestion to explore autoencoders for dimensionality reduction and we plan to investigate this direction in future work.
>
> Q4:
>
> Our method is centered on computing the reachable set of the output space. Once this set is obtained, it can be used to reason about a variety of tasks—not just SSN. While our current paper focuses on challenges posed by high-dimensional outputs in SSN, the proposed approach is general and can be applied to other high-dimensional tasks as well.
>
> Accordingly, we have decided to revise the title if the paper is accepted. The updated title will be:
>
> "Probabilistic Robustness Analysis in High-Dimensional Space: Application to Semantic Segmentation Networks"
>
> Q5:
>
> Yes, our framework provides a principled approach for selecting these parameters. The user specifies a target coverage level $\delta_1$ and confidence level $\delta_2$, and we use a binary search to efficiently determine the appropriate calibration size $m$ and rank $\ell$.
>
> In the common case where $\ell = m - 1$, increasing $m$ monotonically increases $\delta_2$ for a fixed $\delta_1$. This property allows binary search to efficiently identify the optimal $m$ that satisfies the desired level for $\delta_2$. Our toolbox includes this module that retuens optimal $m$ based on desired statistical guarantees.
>
> ---
>
> ***Response to Limitation Comments:***
>
> We would like to again encourage the reviewer to revisit Appendix A, where we provide a detailed explanation of why CI is distribution-free.
>
> You raised three important points, and we address them below:
>
> 1- Importantly, the formal guarantee in our method is entirely derived from CI; the surrogate model is introduced solely to reduce the conservatism of this guarantee, without affecting its validity.
>
> 2- It’s important to emphasize that CI is distribution-free—this is thoroughly discussed in Appendix A. The distribution-free nature of CI is precisely why we adopt it for reachability analysis of neural networks, whose output distributions is complex. Therefore, the validity of our method does not depend on how exact the calibration distribution is captured. As long as the calibration dataset is sampled from the same distribution that guarantee holds and this is the **magic** of CI.
>
> 3- We add a paragraph on societal impact, and the importance of verification in safety-critical domains such as medical imaging.

---

> ### Author Response · Authors · 2025-08-06
> **Request for your feedback**
>
> Dear Reviewer Xbuv
>
> Thank you for your valuable feedback and insightful comments.
>
> We have addressed all of your comments in detail and are currently awaiting your response. As the reviewer-author discussion period is approaching its deadline, we wanted to kindly follow up and ask if you had any feedback on our responses.
>
> Thank you again for your time and consideration.

---

> ### Comment · Reviewer_Xbuv · 2025-08-06
>
> Thank you for the detailed and thoughtful rebuttal. I appreciate the clarifications and the new experiments added to address concerns raised during the initial review.
>
> 1. Your clarification that the ReLU surrogate model is used to tighten the probabilistic reachable sets—rather than to approximate the original SSN precisely—is helpful. The empirical comparison between surrogate-based and naive methods in Appendix E (RQ4), as well as the newly added 10⁶-sample lower bound experiment, is a valuable addition and demonstrates meaningful reductions in conservatism (10–20% reachset inflation). This directly addresses my concern about quantifying conservativeness, and I appreciate this additional effort. However, I still find that some limitations around surrogate fidelity in complex architectures (e.g., very deep or heterogeneous SSNs) are underexplored. While you mention the potential to fall back to mean-based reachability in these cases, more principled discussion (e.g., error bounds, confidence estimates for approximation quality) would strengthen the framework's practical utility.
>
> 2. I understand your rationale for not including PGD/FGSM in the evaluation, especially given their limited dimensionality and the motivation for reachability-based methods in high-dimensional perturbation settings. That said, including at least some standard attack models (even as baselines) would improve alignment with the robustness literature and broaden the appeal and generalizability of the method.
>
> 3. You clarified that “unknown” pixels arise from conservatism in the reachability analysis, and you reference recent experiments showing fewer unknowns compared to other probabilistic techniques. That is useful context. Nevertheless, the practical impact of these regions remains ambiguous—particularly in medical or safety-critical domains where pixel-level robustness matters. Including examples or discussion on downstream decision-making (e.g., alerting users, fallback mechanisms) could help bridge the gap between theoretical guarantees and practical utility.
>
> 4. Your use of deflation-based PCA is both innovative and appropriate for high-dimensional output settings. Still, I encourage future work on exploring alternatives like autoencoders or low-rank factorization, particularly when PCA struggles due to output correlation or class imbalance (as you acknowledged). It’s promising to hear that such directions are under consideration.
>
> In summary, your rebuttal addressed many of my questions with thoughtful explanation and additional experiments. The proposed approach remains a strong contribution in adapting conformal inference to structured, high-dimensional robustness verification, with impressive scalability results. However, I continue to see limitations in the surrogate approximation fidelity, evaluation breadth (e.g., threat models), and interpretation of uncertain outputs that modestly temper the overall impact.
>
> I therefore choose to maintain my original score. Thank you again for the clear and thorough response.

---

> > ### Author Response · Authors · 2025-08-08
> >
> > Dear Reviewer,
> >
> > Thank you for updating your reply.
> >
> > You raised three concerns in your final response, and we would like to address them again to defend our work:
> >
> > 1. The naïve approach is itself one of the contributions of this paper, and it has demonstrated noticeable performance in probabilistic reachability. We also proposed a surrogate-based technique to improve this approach and reduce conservatism. Applying reachability to SSN using the proposed naïve approach is a core contribution of this work, whereas your concern appears to be focused on its improvement, which is presented as a supplementary enhancement.
> >
> > 2. Regarding your first concern that the small surrogate model may not approximate the original model well, our reply is that the ultimate goal in training the small surrogate is to approximate the shape and size of the boundary of the reach set rather than to replicate the original model’s inference. Approximating the reach set is a much easier task, which can be accomplished using smaller surrogates. We do this by lowering the quantile of errors and then performing deterministic reachability on the small surrogate model.
> >
> > 3. The existence of unknown and uncertifiable pixels is a very common issue in the literature. All verification papers—whether deterministic or probabilistic—face this problem by default. The challenge lies in reducing the number of uncertifiable pixels, and this is where methodological improvements become significant. In our view, the mere presence of uncertifiable pixels would be better to not be considered as a reason for borderline-rejection. Like other works, our work also contributes to advancing the verification literature, and the complete elimination of uncertifiability remains an open challenge that the community is actively pursuing.
> >
> > Thanks again for your engagement in this rebuttal and the evaluation of this work.

---

### Note · Authors · 2025-08-13

Dear Area Chairs:

We appreciate the reviewers’ time and effort in evaluating our submission. As all reviewers noted, our paper contributes a scalable technique for the verification of large-scale neural networks. Below is a summary of the reviews:

---

Reviewer PV2T maintained a positive view of the paper, highlighting the contribution in mitigating conservatism in verification of large-scale SSNs. The reviewer suggested including more numerical results, and we addressed this by substantially expanding our experiments.

Reviewer uuuW also had a positive view of the paper. The reviewer was interested in the novelty of our verification approach for SSNs but wanted assurance on how it performs compared to several other techniques (Randomized smoothing) where certify a smoothed approximation of the original model under a worst-case distribution. Whereas our method certifies the original model directly but under assuming a prior distribution. As we both agreed, our approach—with its new guarantee formulation—achieves roughly 100× less conservatism, while acknowledging that the formulations themselves are different. Pleased with the results, the reviewer supported acceptance of this work, noting that the reported accuracy reflects a potential trade-off between techniques rather than a clear advantage, given the differences in guarantee formulations.

Reviewer BgoC also expressed a positive opinion about the paper. He was particularly interested in our integration of a surrogate model and  PCA with CI to reduce the conservatism of our first proposed approach. They agreed that, unlike pixelwise techniques (that suffer from the union bound), our method is scalable to high-dimensional outputs without this limitation. They also noted that the approach can be applied to models where deterministic techniques are not scalable. Additionally, the reviewer suggested including more experimental results on the Cityscapes dataset—a request we have already addressed by adding the comparison that reviewer uuuW asked for.

However, Reviewer Xbuv chose to maintain their original score, though we respectfully disagree, as we have thoroughly explained our reasoning and addressed all concerns.

Our results and comparisons show that this paper proposes a scalable, low-conservatism technique for probabilistic robustness analysis of large-scale semantic segmentation networks, with strong guarantees crucial for vision systems in autonomous driving and medical applications.

Thank you

---

### Decision · Program_Chairs · 2025-09-17

**Decision:**

Accept (poster)

**Comment:**

This paper explores probabilistic verification of the robustness of semantic segmentation networks (SSN). To this end, the paper mainly leverage conformal prediction over reachsets, providing robustness guarantees.


**Strengths**:
* handling an important problem – robustness of SSNs
* novel perspective on certified robustness guarantees – leveraging conformal prediction for probabilistic verification on SSNs
* scalable compared to deterministic verification

**Weaknesses**:
* narrow evaluation
* lack of interpretation on qualitative results

This is a borderline paper but the strengths outweigh the weaknesses, so I and most reviewers  agree for acceptance.

Please update the final manuscript, including, but not limited to, the following promised points:

* The additional results and comparisons to related works (including the extended [b] and [d] table and the Cityscapes/HRNetV2 experiments) are included in the paper.
* The clarifications and additional details from the author responses are integrated, including the distinction between the naive and surrogate methods, and the justification for the chosen threat models.
* The major limitations are explicitly discussed in a limitations section, including (1) the distributional assumption, which may not hold in an adversarial setting, (2) the requirement of recomputing the calibration set for each test point, and (3) the scalability limitations wrt the input dimension of the surrogate approach.